# Subgoal Search For Complex Reasoning Tasks

**Konrad Czechowski***
University of Warsaw
k.czechowski@mimuw.edu.pl

**Tomasz Odrzygóźdź***
University of Warsaw
tomaszo@impan.pl

**Marek Zbysiński**
University of Warsaw
m.zbysinski@students.mimuw.edu.pl

**Michał Zawalski**
University of Warsaw
m.zawalski@uw.edu.pl

**Krzysztof Olejnik**
University of Warsaw
k.olejnik3@student.uw.edu.pl

**Yuhuai Wu**
University of Toronto,
Vector Institute
ywu@cs.toronto.edu

**Łukasz Kuciński**
Polish Academy of Sciences
lkucinski@impan.pl

**Piotr Miłoś**
Polish Academy of Sciences,
University of Oxford,
deepsense.ai
pmilos@impan.pl

## Abstract

Humans excel in solving complex reasoning tasks through a mental process of moving from one idea to a related one. Inspired by this, we propose Subgoal Search (kSubS) method. Its key component is a learned subgoal generator that produces a diversity of subgoals that are both achievable and closer to the solution. Using subgoals reduces the search space and induces a high-level search graph suitable for efficient planning. In this paper, we implement kSubS using a transformer-based subgoal module coupled with the classical best-first search framework. We show that a simple approach of generating $k$-th step ahead subgoals is surprisingly efficient on three challenging domains: two popular puzzle games, Sokoban and the Rubik's Cube, and an inequality proving benchmark INT. kSubS achieves strong results including state-of-the-art on INT within a modest computational budget.

## 1 Introduction

Reasoning is often regarded as a defining property of advanced intelligence [40, 18]. When confronted with a complicated task, humans' thinking process often moves from one idea to a related idea, and the progress is made through milestones, or *subgoals*, rather than through atomic actions that are necessary to transition between subgoals [15]. During this process, thinking about one subgoal can lead to a possibly diverse set of subsequent subgoals that are conceptually reachable and make a promising step towards the problem's solution. This intuitive introspection is backed by neuroscience evidence [20], and in this work, we present an algorithm that mimics this process. Our approach couples a deep learning generative subgoal modeling with classical search algorithms to allow for successful planning with subgoals. We showcase the efficiency of our method on the following complex reasoning tasks: two popular puzzle games Sokoban and the Rubik's Cube, and an inequality theorem proving benchmark INT [55], achieving the state-of-the-art results in INT and competitive results for the remaining two.

The deep learning revolution has brought spectacular advancements in pattern recognition techniques and models. Given the hard nature of reasoning problems, these are natural candidates to provide

*equal contribution

35th Conference on Neural Information Processing Systems (NeurIPS 2021).

search heuristics [4]. Indeed, such a blend can produce impressive results [44, 45, 37, 1]. These approaches seek solutions using elementary actions. Others, e.g. [30, 34, 24], utilize variational subgoals generators to deal with long-horizon visual tasks. We show that these ideas can be pushed further to provide algorithms capable of dealing with combinatorial complexity.

We present Subgoal Search (kSubS) method and give its practical implementations: MCTS-kSubS and BF-kSubS. kSubS consists of the following four components: planner, subgoal generator, a low-level policy, and a value function. The planner is used to search over the graph induced by the subgoal generator and is guided by the value function. The role of the low-level policy is to prune the search tree as well as to transition between subgoals. In this paper, we assume that the generator predicts subgoals that are $k$ step ahead (towards the solution) from the current state, and to emphasize this we henceforth add $k$ to the method's abbreviation. MCTS-kSubS and BF-kSubS differ in the choice of the search engine: the former uses Monte-Carlo Tree Search (MCTS), while the latter is backed by Best-First Search (BestFS). We provide two sets of implementations for the generator, the low-level policy, and the value functions. The first one uses transformer architecture [49] for each component, while the second utilizes a convolutional network for the generator and the value function, and the classical breadth-first search for the low-level policy. This lets us showcase the versatility and effectiveness of the approach.

The subgoal generator lies at the very heart of Subgoal Search, being an implementation of reasoning with high-level ideas. To be useful in a broad spectrum of contexts, the generator should be implemented as a learnable (generative) model. As a result, it is expected to be imperfect and (sometimes) generate incorrect predictions, which may turn the search procedure invalid. Can we thus make planning with learned subgoals work? In this paper, we answer this question affirmatively: we show that the autoregressive framework of transformer-based neural network architecture [49] leads to superior results in challenging domains.

We train the transformer with the objective to predict the $k$-th step ahead. The main advantages of this subgoal objective are simplicity and empirical efficiency. We used expert data to generate labels for supervised training. When offline datasets are available, which is the case for the environments considered in this paper[2], such an approach allows for stable and efficient optimization with high-quality gradients. Consequently, this method is often taken when dealing with complex domains (see e.g. [42, 52]) or when only an offline expert is available[3]. Furthermore, we found evidence of out-of-distribution generalization.

Finally, we formulate the following hypothesis aiming to shed some light on why kSubS is successful: we speculate that subgoal generation may alleviate errors in the value function estimation. Planning methods based on learning, including kSubS, typically use imperfect value function-based information to guide the search. While traditional low-level search methods are susceptible to local noise, subgoal generation allows for evaluations of the value functions at temporally distant subgoals, which improves the signal-to-noise ratio and allows a "leap over" the noise.

To sum up, our contributions are:

1. We propose Subgoal Search method with two implementations: MCTS-kSubS, BF-kSubS. We demonstrate that our approach requires a relatively little search or, equivalently, is able to handle bigger problems. We also observe evidence of out-of-distribution generalization.
2. We provide evidence that a transformer-based autoregressive model learned with a simple supervised objective to predict states $k$-th step ahead is an effective tool to generate valid and diverse subgoals.
3. We show in our experiments that using subgoal planning help to might mitigate the negative influence of value function errors on planning.

We provide the code of our method and experiment settings at `https://github.com/subgoal-search/subgoal-search`, and a dedicated website `https://sites.google.com/view/subgoal-search`.

[2] The dataset for INT or Sokoban can be easily generated or are publicly available. For the Rubik's Cube, we use random data or simple heuristic (random data are often sufficient for robotic tasks and navigation.)

[3] For example, the INT engine can easily generate multiple proves of random statements, but *cannot* prove a given theorem.

## 2 Related work

In classical AI, reasoning is often achieved by *search* ([40]). Search rarely can be exhaustive, and a large body of algorithms and heuristics has been developed over the years, [40, Section 3.5]. It is hypothesized that progress can be achieved by combining search with learning [4]. Among notable successful examples of this approach are Alpha Zero [43], or solving Rubik's cube using a learned heuristic function to guide the $A^*$ algorithm (see [1]).

An eminent early example of using goal-directed reasoning is the PARADISE algorithm ([53]). In deep learning literature, [25] was perhaps the first work implementing subgoal planning. This was followed by a large body of work on planning with subgoals in the latent space for visual tasks ([24, 31, 34, 30, 21, 9, 35]) or landmark-based navigation methods ([41, 27, 14, 46, 56, 23]).

The tasks considered in the aforementioned studies are often quite forgiving when it comes to small errors in the subgoal generation. This can be contrasted with complex reasoning domains, in which even a small variation of a state may drastically change its meaning or render it invalid. Thus, neural networks may struggle to generate semantically valid states ([4, Section 6.1]).

Assuming that this problem was solved, a generated subgoal still remains to be assessed. The exact evaluation may, in general, require exhaustive search or access to an oracle (in which case the original problem is essentially solved). Consequently, it is unlikely that a simple planner (e.g., one unrolling independent sequences of subgoals [9]) will either produce an informative outcome, could be easily improved using only local changes via gradient descent [31], or cross-entropy method (CEM) [30, 34, 35]. Existing approaches, which are based on more powerful subgoal search methods, have their limitations, on the other hand. [13] is perhaps the closest to our method and uses MCTS to search the subgoal-induced graph. However, it uses a predefined (not learned) predicate function as a subgoal generator, limiting applicability to the problems with available high-quality heuristics. Learned subgoals are used in [32], a method of hierarchical planning. That said, the subgoal space needs to be relatively small for this method to work (or crafted manually to reduce cardinality). To the best of our knowledge, our algorithm is the first domain agnostic hierarchical planner for combinatorially complex domains.

More generally, concepts related to goals and subgoals percolated to reinforcement learning early on, leading, among others, to prominent ideas like hindsight [22], hierarchical learning [48, 7] or the Horde architecture [47]. Recently, with the advent of *deep* reinforcement learning, these ideas have been resurfacing and scaled up to deliver their initial promises. For example, [51] implements ideas of [7] and a very successful hindsight technique [2] is already considered to be a core RL method. Further, a recent paper [36] utilizes a maximum entropy objective to select achievable goals in hindsight training and [50] uses meta-learned subgoals to discover options in multi-task RL settings. Apart from the aforementioned hierarchical planning, the idea to use neural networks for subgoal generation was used to improve model-free reinforcement learning agents [11, 6] and imitation learning algorithms [33].

## 3 Method

Our method, Subgoal Search (kSubS), is designed for problems, which can be formulated as a search over a graph with a known transition model. Formally, let $G = (\mathcal{S}, \mathcal{E})$ be a directed graph and $\tilde{\mathcal{S}} \subset \mathcal{S}$ be the set of success states. We assume that, during the solving process, the algorithm can, for a given node $g$, determine the edges starting at $g$ and check if $g \in \tilde{S}$.

Subgoal Search consists of four components: planner, subgoal generator, low-level policy, and a value function. The planner, coupled with a value function, is used to search over the graph induced by the subgoal generator. Namely, for each selected subgoal, the generator allows for sampling the candidates for the next subgoals. Only these reachable by the low-level policy are used. The procedure continues until the solution is found or the computational budget is exhausted. Our method searches over a graph $\tilde{G} = (\mathcal{S}, \mathcal{E}_s)$, with the edges $\mathcal{E}_s$ given by the subgoal generator. Provided a reasonable generator, paths to $\tilde{\mathcal{S}}$ are shorter in $\tilde{G}$ than in $G$ and thus easier to find.

In this paper we provide BestFS- and MCTS- backed implementations kSubS. Algorithm 1 presents BF-kSubS; see Algorithm 4 in Appendix A.1 for MCTS-kSubS.

For INT and Rubik's Cube, we use transformer models (see Appendix B.1) in all components other than the planner. For Sokoban, we use convolutional networks (see Appendix B.2). While transformers could also be used in Sokoban, we show that a simplified setup already achieves strong results. This showcases that Subgoal Search is general enough to work with different design choices. In Section 4.2 we describe datasets used train these neural models.

**Algorithm 1** Best-First Subgoal Search (BF-kSubS)

```
Require:  C_1      max number of nodes
          V        value function network
          SOLVED   predicate of solution

function SOLVE(s_0)
    T.PUSH((V(s_0), s_0))          ▷ T is priority queue
    paths[s_0] ← []                ▷ paths is dict of lists
    seen.ADD(s_0)                  ▷ seen is set
    while 0 < LEN(T) and LEN(seen) < C_1 do
        _, s ← T.EXTRACT_MAX()
        subgoals ← SUB_GENERATE(s)
                                   ▷ see Algorithm 3
        for s' in subgoals do
            if s' in seen then continue
            seen.ADD(s')
            actions ← GET_PATH(s, s')
                                   ▷ see Alg 2 or Alg 9
            if actions.EMPTY() then
                continue
            T.PUSH((V(s'), s'))
            paths[s'] ← paths[s] + actions
            if SOLVED(s') then
                return paths[s']
    return False
```

**Algorithm 2** Low-level conditional policy

```
Require:  C_2   steps limit
          π     low-level conditional
                policy network
          M     model of the environment

function GET_PATH(s_0, subgoal)
    step ← 0
    s ← s_0
    action_path ← []
    while step < C_2 do
        action ← π.PREDICT(s, subgoal)
        action_path.APPEND(action)
        s ← M.NEXT_STATE(s, action)
        if s = subgoal then
            return action_path
        step ← step + 1
    return []
```

**Subgoal generator** Formally, it is a mapping $\rho\colon \mathcal{S} \to \mathbf{P}(\mathcal{S})$, where $\mathbf{P}(\mathcal{S})$ is a family of probability distributions over the environment's state space $\mathcal{S}$. More precisely, let us define a trajectory as a sequence of state-action pairs $(s_0, a_0), (s_1, a_1), \ldots, (s_n, a_n)$, with $(s_i, a_i) \in \mathcal{S} \times \mathcal{A}$, where $\mathcal{A}$ stands for the action space and $n$ is the trajectory's length. We will say that the generator predicts $k$*-step ahead* subgoals, if at any state $s_\ell$ it aims to predict $s_{\min(\ell+k,n)}$. We show, perhaps surprisingly, that this simple objective is an efficient way to improve planning, even for small values of $k$, i.e. $k \in \{2, 3, 4, 5\}$.

Operationally, the subgoal generator takes as input an element of the **s** $\in \mathcal{S}$ and returns *subgoals*, a set of new candidate states expected to be closer to the solution and is implemented by Algorithm 3. It works as follows: first, we generate $C_3$ subgoal candidates with SUB_NET_GENERATE. Then, we prune this set to obtain a total probability greater than $C_4$.

For INT and Rubik's Cube, we represent states as sequences modeled with a transformer. Following the practice routinely used in language modeling, [49], SUB_NET_GENERATE employs beam search to generate a set of high likelihood outputs.[4] With the same goal in mind, for Sokoban, we use another method described Appendix C.1. SUB_NET_GENERATE uses the subgoal generator network, which is trained in a supervised way on the expert data, with training examples being: $s_\ell$ (input) and $s_{\min(\ell+k,n)}$ (label), see Appendix D for details.

**Low-level conditional policy** Formally, it is a mapping $\pi\colon \mathcal{S} \times \mathcal{S} \to \mathcal{A}^*$. It is used to generate a sequence of actions on how to reach a subgoal starting from a given initial state. Operationally, it may return an empty sequence if the subgoal cannot be reached within $C_2$ steps, see Algorithm 2. This is used as a pruning mechanism for the *subgoals* set in Algorithm 1.

[4]In language modeling, typically, only one beam search output is used. In our case, however, we utilize all of them, which turns out to be a diverse set of subgoals.

In INT and Rubik's Cube, we use Algorithm 2, which utilizes low-level policy network $\pi$. Similarly to the subgoal generator, it is trained using expert data in a supervised fashion, i.e. for a pair $(s_\ell, s_{\min(\ell+i,n)})$, with $i \leq k$, its objective is to predict $a_\ell$.

When the branching factor is small, the low-level policy can be realized by a simple breadth-first search mechanism, see Algorithm 9 in Appendix, which we illustrate on Sokoban.

**Algorithm 3** Subgoal generator

```
Require:  C_3   number of subgoals
          C_4   target probability
          ρ     subgoal generator network

function SUB_GENERATE(s)
    subgoals ← ∅
    states, probs ← SUB_NET_GENERATE(ρ, s; C_3)
                ▷ (states, probs) is sorted wrt probs
    total_p ← 0
    for state, p ∈ (states, probs) do
        if total_p > C_4 then break
        subgoals.ADD(state)
        total_p ← total_p + p
    return subgoals
```

**Value function** Formally, it is a mapping $V: \mathcal{S} \to \mathbb{R}$, that assigns to each state a value related to its distance to the solution, and it is used to guide the search (see Algorithm 1 and Algorithm 4). For its training, we use expert data. For each state $s_\ell$ the training target is negated distance to the goal: $\ell - n$, where $n$ denotes the end step of a trajectory that $s_\ell$ belongs to.

**Planner** This is the engine that we use to search the subgoal-induced graph. In this paper, we use BestFS (Algorithm 1) and MCTS (Algorithm 4 in Appendix A.1). The former is a classic planning method, which maintains a priority queue of states waiting to be explored, and greedily (with respect to their value) selects elements from it (see, e.g., [40]). MCTS is a search method that iteratively and explicitly builds a search tree, using (and updating) the collected node statistics (see, e.g., [5]). In this paper, we use an AlphaZero-like [42] algorithm for single-player games.

We note that the subgoal generator can be combined with virtually any search algorithm and can benefit from an additional structure. For example, for domains providing a factored state representation, the width-based methods [26, 12] would likely be stronger search mechanisms.

## 4 Experiments

In this section, we empirically demonstrate the efficiency of MCTS-kSubS and BF-kSubS. In particular, we show that they vastly outperform their standard ("non-subgoal") counterparts. As a testing ground, we consider three challenging domains: Sokoban, Rubik's Cube, and INT. All of them require non-trivial reasoning. The Rubik's Cube is a well-known 3-D combination puzzle. Sokoban is a complex video puzzle game known to be NP-hard and thus challenging for planning methods. INT [55] is a recent theorem proving benchmark.

### 4.1 Training protocol and baselines

Our experiments consist of three stages. First, we collect domain-specific expert data, see Section 4.2. Secondly, we train the subgoal generator, low-level conditional policy, and value function networks using the data and targets described in Section 3. For more details see Appendix D. Eventually, we evaluate the planning performance of MCTS-kSubS and BF-kSubS, details of which are presented below. In the second step, we use supervised learning, which makes our setup stable with respect to network initialization, see details in Appendix D.1.3.

As baselines, we use BestFS and MCTS (being a single-player implementation of AlphaZero). In INT and Rubik's Cube, both the algorithms utilize policy networks (trained with behavioral cloning, on the same dataset, which we used to train kSubS). Note that distribution over actions induces a distribution over states; thus the policy network can be regarded as a subgoal generator for $k = 1$. More details about the baselines can be found in Appendix I.

### 4.2 Search Domains and Datasets

**Sokoban** is a single-player complex game in which a player controls an agent whose goal is to place boxes on target locations solely by pushing them; without crossing any obstacles or walls. Sokoban has recently been used to test the boundaries in RL [16, 29]. Sokoban is known to be hard [10], mainly due to its combinatorial complexity and the existence of irreversible states. Deciding if a given Sokoban board is solvable is an NP-hard problem [8].

We collect the expert dataset consisting of all successful trajectories occurring during the training of an MCTS agent (using an implementation of [29]). These are suboptimal, especially in the early phases of the training or for harder boards. For both expert training and kSubS evaluation, we generate Sokoban boards following the approach of [38].

**Rubik's Cube** is a classical 3-dimensional puzzle. It is considered challenging due to the fact that the search space has more than $4.3 \times 10^{18}$ configurations. Similarly to Sokoban, Rubik's Cube has been recently used as a testing ground for RL methods [1].

To collect the expert dataset, we generate random paths of length 30 starting from the solved cube and take them backward. These backward solutions are highly sub-optimal (optimal solutions are proven to be shorter than 26 [39]).

**INT: Inequality Benchmark for Theorem Proving**. INT provides a generator of inequalities, which produces mathematical formulas along with their proofs, see [55, Section 3.3]. Proofs are represented as sequences of consecutively applied mathematical axioms (there are $K = 18$ axioms in total). An action in INT is a tuple containing an axiom and its input entities. The action space in this problem can be very large, reaching up to $10^6$ elements, which significantly complicates planning.

The INT generator constructs paths by randomly applying axioms starting with a trivial statement. Such a path taken backward constitutes the proof of its final statement (not guaranteed to be optimal). The proof length, denoted by $L$, is an important hyperparameter regulating the difficulty – we use $5, 10, 15$.

For more details on datasets see Appendix C.

| | INT | Sokoban | Rubik |
|---|---|---|---|
| $k$ | 3 | 4 | 4 |
| $C_1$ | 400 | 5000 | 1500 |
| $C_2$ | 4 | 4 | 7 |
| $C_3$ | 4 | 4 | 3 |
| $C_4$ | 1 | 0.98 | 1 |

Table 1: BF-kSubS hyperparameters.

### 4.3 Main results

In this section, we present our most important finding: Subgoal Search enables for more efficient search and consequently scales up to problems with higher difficulty. Specifically, MCTS-kSubS and BF-kSubS perform significantly better than the respective methods not using subgoals, including state-of-the-art on INT.

In Figure 1, we present the performance of Subgoal Search. We measure the *success rate* as a function of the *search budget*. The success rate is measured on 1000 instances of a given problem (which results in confidence intervals within $\pm 0.03$). For BF-kSubS the search budget is referred to as *graph size* and includes the number of nodes visited by Algorithm 1. For INT and Rubik's Cube, we include both the subgoal generated by `SUB_GENERATE` and the nodes visited by `GET_PATH` (as they induce a significant computational cost stemming from using low-level policy $\pi$ in Algorithm 2). For Sokoban, we use Algorithm 9 to realize `GET_PATH`, as it has a negligible cost (less than $1\%$ of the total runtime of Algorithm 1), we do not include these nodes into graph size.

For MCTS, we report *MCTS passes*, which is a common metric for MCTS, see details in Appendix A.1.

Below we discuss the results separately for each domain. We provide examples of solutions and generated subgoals in Appendix H.

**INT** The difficulty of the problems in INT increases fast with the proof length $L$ and the number of accessible axioms. W used $K = 18$; all of available axioms. We observe, that BF-kSubS scales to proofs of length $L = 10$ and $L = 15$, which are significantly harder than $L = 5$ considered in [55], see Table 2. The same holds for MCTS-kSubS, see Appendix A.2.

We check also that MCTS-kSubS vastly outperforms the baseline - AlphaZero algorithm, see Figure 1 (top, left). An MCTS-based agent was also evaluated in [55]. Its implementation uses graph neural

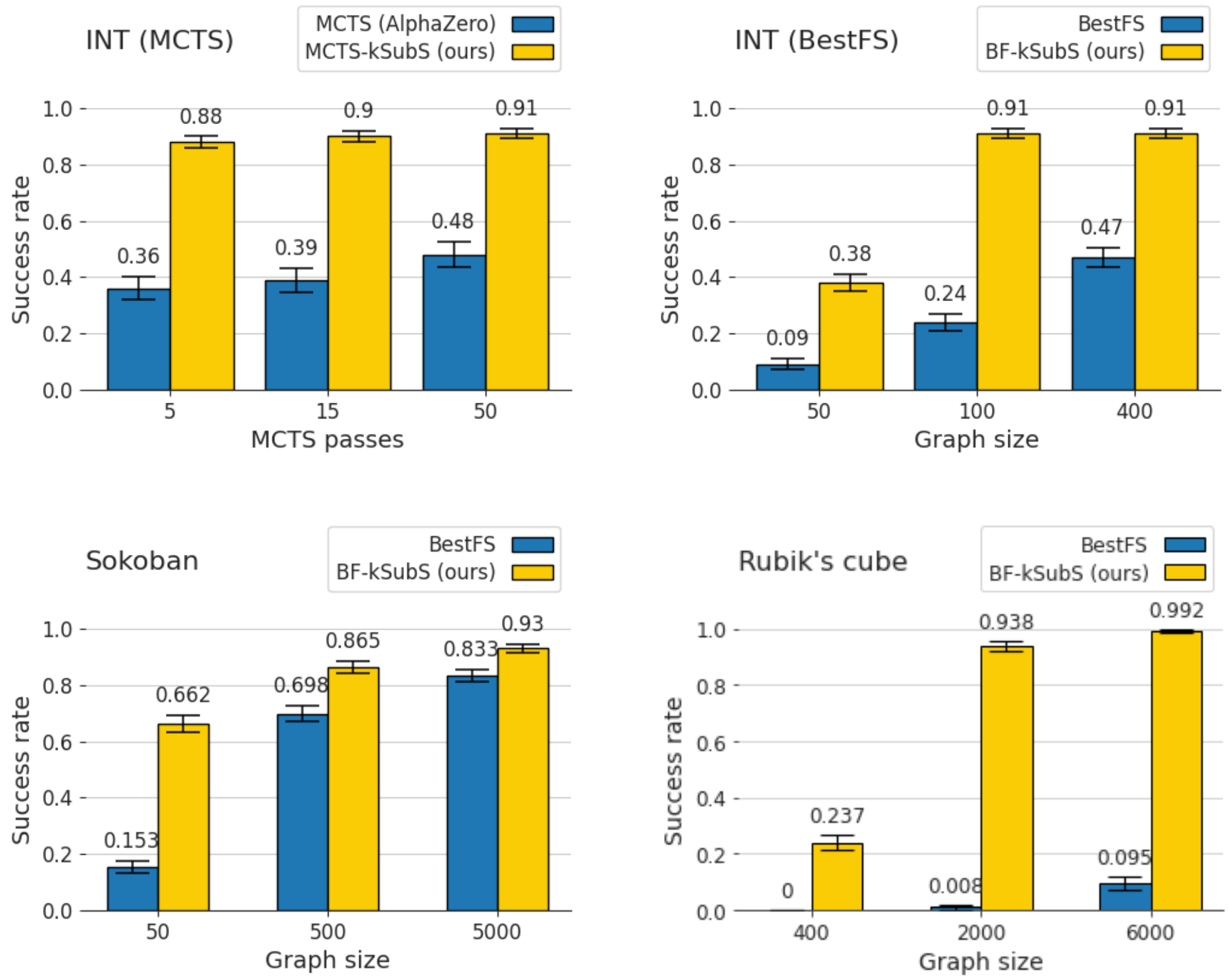

Figure 1: The performance of Subgoal Search. (top, left) comparison on INT (with the proof length 15) to AlphaZero. (top, right) BF-kSubS consistently achieves high performance even for small computational budgets. (bottom, left) similarly on Sokoban (board size 12x12 with 4 boxes) the advantage of BF-kSubS is clearly visible for small budget. (bottom, right) BestFS fails to solve Rubik's Cube, while BF-kSubS can achieve near-perfect performance.

| | Proof length | 5 | | 10 | | 15 | |
|---|---|---|---|---|---|---|---|
| | Method | BestFS | BF-kSubS (ours) | BestFS | BF-kSubS (ours) | BestFS | BF-kSubS (ours) |
| Graph size | 50 | 0.82 | **0.99** | 0.47 | **0.97** | 0.09 | **0.38** |
| | 100 | 0.89 | **0.99** | 0.64 | **0.99** | 0.24 | **0.91** |
| | 200 | 0.92 | **0.99** | 0.67 | **0.99** | 0.35 | **0.91** |
| | 400 | 0.93 | **0.99** | 0.72 | **0.99** | 0.47 | **0.91** |

Table 2: INT success rates for various proof lengths and graphs sizes.

networks architectures and achieves $92\%$ success rate for $L = 5$. Our transformed-based baseline is stronger - it solves over $99\%$ instances on the same problem set.

**Sokoban** Using BF-kSubS allows for significantly higher success rates rates within the same computational budget, see Table 3. Our solution scales well to the board size as big as $20 \times 20$; note that $10 \times 10$ boards are typically used in deep RL research [16, 38]. Importantly, we observe that already for a small computational budget (graph size 1000) BF-kSubS obtains higher success rates than the expert we used to create the dataset (these are $78\%$, $67\%$, $60\%$ for the respective board sizes).

We also tested how the quality of BF-kSubS depends on the size of the training dataset for Sokoban, the results can be found in Appendix F.

**Rubik's Cube** BF-kSubS solves nearly $100\%$ of cubes, BestFS solve less than $10\%$, see Figure 1 (bottom, right). This is perhaps the most striking example of the advantage of using a subgoal generator instead of low-level actions. We present possible explanation in Appendix K.

| | Board size | 12 x 12 | | 16 x 16 | | 20 x 20 | |
|---|---|---|---|---|---|---|---|
| | Method | BestFS | BF-kSubS (ours) | BestFS | BF-kSubS (ours) | BestFS | BF-kSubS (ours) |
| Graph size | 50 | 0.15 | **0.66** | 0.04 | **0.42** | 0.02 | **0.46** |
| | 100 | 0.46 | **0.79** | 0.23 | **0.62** | 0.10 | **0.55** |
| | 1000 | 0.75 | **0.89** | 0.61 | **0.79** | 0.47 | **0.70** |
| | 5000 | 0.83 | **0.93** | 0.69 | **0.85** | 0.58 | **0.77** |

Table 3: Sokoban success rates for various board sizes (each with 4 boxes).

**Out-of-distribution (OOD) generalization** OOD generalization is considered to be the crucial ability to make progress in hard combinatorial optimization problems [4] and automated theorem proving [55]. The INT inequality generator has been specifically designed to benchmark this phenomenon. We check that Subgoal Search trained on proofs on length 10 generalizes favourably to longer problems, see Figure 4. Following [55], we speculate that search is a computational mechanism that delivers OOD generalization.

It might be hard to compare computational budgets between various algorithms and their versions. In Appendix E we measure that BF-kSubS and MCTS-kSubS offer very practical benefits, sometimes as much as $7\times$ faster execution.

### 4.4 Analysis of $k$ (subgoal distance) parameter

The subgoals are trained to predict states $k$ steps ahead of the current one. Higher $k$ should make planning easier as the search graph is smaller. However, as $k$ increases, the quality of the generator may drop, and thus the overall effect is uncertain. Similarly, the task of the low-level conditional policy becomes more difficult as $k$ increases. The optimal value of $k$ is 3 and 4 for INT and Rubik's Cube, respectively. In these environments, increasing $k$ further degrades performance. In Sokoban, we observe monotonic improvement up to $k = 10$. This is perhaps because low-level conditional policy (Algorithm 9, based on breadth-first search) never fails to fill the path from a state to the valid subgoal. The running cost of Algorithm 9 quickly becomes unacceptable (recall that for $k = 4$, which we used in the main experiment, it has still a negligible cost - below $< 1\%$ of the total runtime).

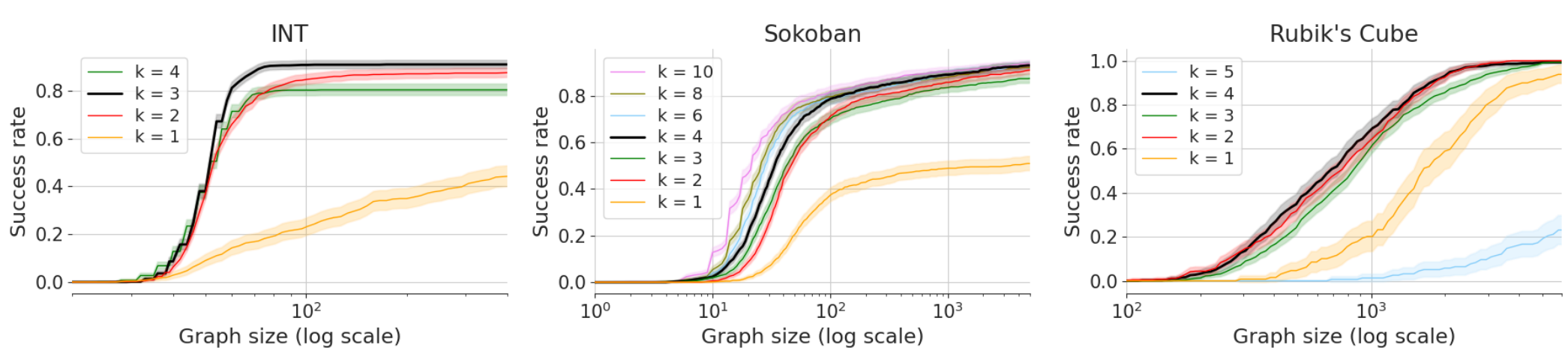

Figure 2: BF-kSubS success rates for different values of $k$. Black curves represent the values of $k$ used in the main experiments (that is $k = 4$ for Rubik's Cube and Sokoban and $k = 3$ for INT).

### 4.5 Quality of subgoals

The learned subgoal generator is likely to be imperfect (especially in hard problems). We study this on $10 \times 10$ boards of Sokoban, which are small enough to calculate the true distance $dist$ to the solution using the Dijkstra algorithm. In Figure 3, we study $\Delta := dist_{s_1} - dist_{s_2}$, where $s_1$ is a sampled state and $s_2$ is a subgoal generated from $s_1$. Ideally, the histogram should concentrate on $k = 4$ used in training. We see that in slightly more than $65\%$ of cases subgoals lead to an improvement.

The low-level conditional policy in Algorithm 2 provides additional verification of generated states. We check that in INT and Rubik's Cube, about $50\%$ of generated subgoals can be reached by this policy (the rest is discarded).

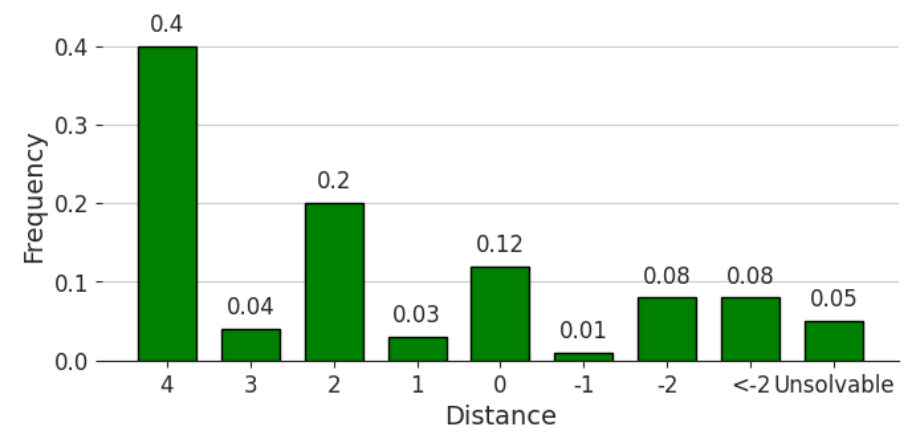

Figure 3: Histogram of $\Delta$. Note that 17% of subgoals increases the distance. Additional, 5% leads to unsolvable “dead states” present in Sokoban.

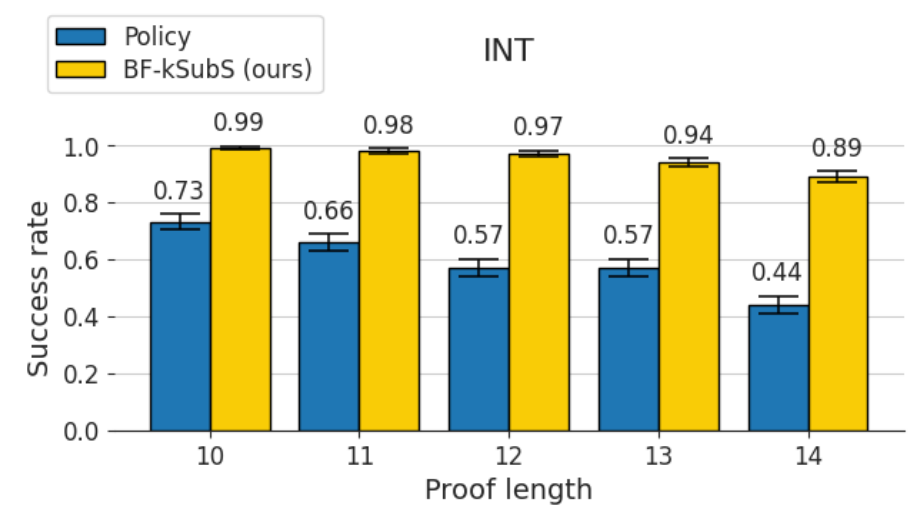

Figure 4: Out-of-distribution generalization to longer proofs. We compare with the behavioral cloning agent (Policy) studied in [55].

### 4.6 Value errors

There might be various explanations for the success of our method. One of them is that Subgoal Search better handles errors of learned value functions. In this section, we discuss this using a synthetic grid world example and performing statistical analysis on the real value function trained to approximate the distance to the solution (as described in Section 3).

**Grid world example** Consider a grid world with the state space $S = \{1, \ldots, n\}^m$, with $(0, \ldots, 0)$ being the initial state and $(n, \ldots, n)$ the goal state. A pair of states is connected by an edge if they are at distance 1 from each other. Let:

- Synthetic value function: the negative distance to the goal plus i.i.d Gaussian noise $\mathcal{N}(0, \sigma^2)$.
- Synthetic SUB_GENERATE (instead of Algorithm 3): Let $B_k(s)$ be the states within distance $k$ from $s$. We return $C_3 - 1$ states sampled uniformly from $B_k(s)$ and a one "good subgoal" being a state in $B_k(s)$ with the minimal distance to the solution.
- Node expansion in BestFS (baseline): implemented as SUB_GENERATE above with $k = 1$.

In this setup, one easily sees that the probability that the good subgoal will have the highest value estimation among the generated states grows with $k$. Consequently, kSubS can handle higher levels of noise than the baseline BestFS, see Table 4.

| $\sigma$ | BestFS | BF-kSubS |
|---|---|---|
| 3 | 0.999 | 1 |
| 10 | 0.142 | 1 |
| 20 | 0.006 | 0.983 |

Table 4: Success rates on the grid world ($m = 6, n = 10$), depending on the value function noise scale. We use the search budget of 500 nodes and $k = 4$ for kSubS.

**Value monotonicity** Imagine a solution path from the starting state to the goal state. Due to the errors, the value estimates on the path may not be monotonic. This is an undesirable property, which is likely to hinder search and make finding the path harder. Now consider the subpath consisting of consecutive states spaced $k$ actions apart, as can be constructed by Subgoal Search. For this version, the value estimates are more likely to be monotonic and easier to find. To illustrate this, we measure monotonicity on solution paths found by our algorithm for INT. The probability that value decreases when moving $k$ steps ahead drops from $0.32$ when $k = 1$ to mere $0.02$ for $k = 4$ (see Table 7 in Appendix).

**Overoptimism** Alternatively, one can consider that erroneously positive values misguide a search method (a phenomenon known as over-optimism [19]). To illustrate this, consider $\mathcal{S}_3(s)$, the set of all states having the same distance to the solution as $s$ and within distance 3 from $s$. Intuitively, $\mathcal{S}_3(s)$ contains similar states with respect to the difficulty of solving. In Sokoban, the standard deviation of value function prediction for $\mathcal{S}_3(s)$ is equal to $2.43$ (averaged over different $s$ on Sokoban boards). This is high when compared to the average increase of value for moving one step closer to the solution, which is only $1.34$. Consequently, it is likely that $\mathcal{S}_3(s)$ contains a suboptimal state, e.g., having a higher value than the best immediate neighbors of $s$ (which by properties of the game will be closer to solution in Sokoban). Indeed, we measure that the probability of such an event is $64\%$. However, it drops significantly to $29\%$ if one considers states closer by $4$ steps (say given by a subgoal generator).

## 5 Limitations and future work

In this section, we list some limitations of our work and suggest further research directions.

**Reliance on expert data** In this version, we use expert data to train learnable models. As kSubS improves the performance, we speculate that training akin to AlphaZero can be used, i.e. in a planner-learner loop without any outside knowledge.

**Optimality and completeness** kSubS searches over a reduced state space, which might produce suboptimal solutions or even fail to find them. This is arguably unavoidable if we seek an efficient method for complex problems.

**Subgoals definition** We use simple $k$-step ahead subgoals, which is perhaps not always optimal. Our method can be coupled with other subgoal paradigms. Unsupervised detection of landmarks (see e.g. [56]) seems an attractive future research direction.

**More environments** In future work, we plan to test kSubS on more environments to understand its strengths and weaknesses better. In this work, we generate subgoals in the state space, which might be limiting for tasks with high dimensional input (e.g., visual).

**Reliance on a model of the environment** We use a perfect model of the environment, which is a common practice for some environments, e.g., INT. Extending kSubS to use learned (imperfect) models is an important future research direction.

**Determinism** Our method requires the environment to be deterministic.

**OOD generalization** A promising future direction is to investigate and leverage the out-of-distribution generalization delivered by our method and compare to (somewhat contradictory) findings of [17, 55].

**Classical planning methods** For many search problems, the state space can be represented in factored fashion (or such representation can be learned [3]). In such cases, the search can be greatly improved with width-based methods [26, 12]. It is an interesting research direction to combine kSubS with such methods.

## 6 Conclusions

We propose Subgoal Search, a search algorithm based on subgoal generator. We present two practical implementations MCTS-kSubS and BF-kSubS meant to be effective in complex domains requiring reasoning. We confirm that indeed our implementations excel in Sokoban, Rubik's Cube, and inequality benchmark INT. Interestingly, a simple $k$ step ahead mechanism of generating subgoals backed up by transformer-based architectures performs surprisingly well. This evidence, let us hypothesize, that our methods (and related) can be further scaled up to even harder reasoning tasks.

## Acknowledgments and Disclosure of Funding

The work of Konrad Czechowski, Tomasz Odrzygóźdź and Piotr Miłoś was supported by the Polish National Science Center grant UMO-2017/26/E/ST6/00622. This research was supported by the PL-Grid Infrastructure. Our experiments were managed using `https://neptune.ai`. We would like to thank the Neptune team for providing us access to the team version and technical support.

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
