# A MCTS

## A.1 MCTS-kSubS algorithm

In Algorithm 4 we present a general MCTS solver based on AlphaZero. Solver repeatedly queries the planner for a list of actions and executes them one by one. Baseline planner returns only a single action at a time, whereas MCTS-kSubS gives around $k$ actions – to reach the desired subgoal (number of actions depends on a subgoal distance, which not always equals $k$ in practice).

MCTS-kSubS operates on a high-level subgoal graph: nodes are subgoals proposed by the generator (see Algorithm 3) and edges – lists of actions informing how to move from one subgoal to another (computed by the low-level conditional policy in Algorithm 2). The graph structure is represented by $tree$ variable. For every subgoal, it keeps up to $C_3$ best nearby subgoals (according to generator scores) along with a mentioned list of actions and sum of rewards to obtain while moving from the parent to the child subgoal.

Most of MCTS implementation is shared between MCTS-kSubS and AlphaZero baseline, as we can treat the behavioral-cloning policy as a subgoal generator with $k = 1$. All the differences between MCTS-kSubS and the baseline are encapsulated in GEN_CHILDREN function (Algorithms 5 and 6). To generate children subgoals MCTS-kSubS runs subgoal generator and low-level conditional policy, whereas the baseline uses behavioral cloning policy for that purpose.

---

**Algorithm 4** MCTS solver (common for AlphaZero baseline and MCTS-kSubS)

---

**Require:**
| | | |
|---|---|---|
| | $L_a$ | action limit |
| | $L_p$ | planner calls limit |
| | $P$ | planning passes |
| | $\gamma$ | discount factor |
| | $c_{puct}$ | exploration weight |
| | $\tau$ | sampling temperature |
| | $V$ | value function |
| | $env$ | environment |
| | $M$ | environment model |

**Use:**
| | | |
|---|---|---|
| | $tree$ | tree structure |
| | $N(s,i)$ | visit count |
| | $W(s,i)$ | total child-value |
| | $Q(s,i)$ | mean child-value |
| | $\pi_e$ | exploration policy |

```
# Initialize N, W, Q to zero
function SOLVER
    s ← env.RESET()
    solution ← []          ▷ List of actions
    for 1 ... L_p do
        actions ← PLANNER(s)
        for a in actions do
            s', r ← env.STEP(a)
            solution.APPEND(a)
            s ← s'
        if solution.LENGTH() > L_a then
            return None
        if env.SOLVED() then
            return solution
    return None

function PLANNER(state)
    for 1 ... P do
        path, leaf ← SELECT(state)
        EXPAND(leaf)
        UPDATE(path, leaf)
    return CHOOSE_ACTIONS(state)
```

```
function SELECT(state)
    s ← state
    path ← []
    while s belongs to tree do
        i ← SELECT_CHILD(s)
        s', r, actions ← tree[s][i]
        path.APPEND((s, i, r))
        s ← s'
    return path, s

function EXPAND(leaf)
    children, probs ← GEN_CHILDREN(leaf)
    tree[leaf] ← children
    π(leaf, ·) ← probs
    for i ← 1 to children.LENGTH() do
        s', r, actions ← tree[leaf][i]
        W(leaf, i) ← r + γ * V(s')
        N(leaf, i) ← 1
        Q(leaf, i) ← W(leaf, i)

function UPDATE(path, leaf)
    quality ← V(leaf)
    for s, i, r ← reversed(path) do
        quality ← r + γ * quality
        W(s, i) ← W(s, i) + quality
        N(s, i) ← N(s, i) + 1
        Q(s, i) ← W(s,i)/N(s,i)

function SELECT_CHILD(s)
    U(s, i) ← √(Σ_i' N(s, i'))/(1 + N(s, i))
    i ← argmax_i(Q(s, i) + c_puct π_e(s, i)U(s, i))
    return i

function CHOOSE_ACTIONS(s)
    i ∼ softmax(1/τ log N(s, ·))
    s', r, actions ← tree[s][i]
    return actions
```

| **Algorithm 5** GEN_CHILDREN for MCTS-kSubS | **Algorithm 6** GEN_CHILDREN for AlphaZero |
|---|---|

*For functions GET_PATH and SUB_GENERATE see Algorithms 2 and 3.*

```
function GEN_CHILDREN(state)
    s ← state
    children ← []
    probs ← []
    for subgoal, prob ← SUB_GENERATE(s) do
        actions ← GET_PATH(s, subgoal)
        if actions.EMPTY( ) then continue
        r ← M.REWARD_SUM(s, actions)
        children.APPEND((subgoal, r, actions))
        probs.APPEND(prob)
    return children, probs
```

**Require:**    $\pi_b$    behavioral cloning policy
```
function GEN_CHILDREN(state)
    s ← state
    children ← []
    probs ← []
    for a, prob ← π_b.GEN_ACTIONS(s) do
        s', r ← M.NEXT_STATE_REWARD(s, a)
        children.APPEND((s', r, [a]))
        probs.APPEND(prob)
    return children, probs
```

Variables $tree, N, W, Q, \pi_e$ are reused across subsequent planner invocations within a single solver run. We limit the number of planner calls $L_p$ for better control over the computational budget for MCTS-kSubS. For MCTS pseudocode we assume a slightly modified version of SUB_GENERATE function (defined originally in Algorithm 3). We presume that the function along with subgoals returns also their respective probabilities – as MCTS needs them to guide exploration.

## A.2   Detailed results of MCTS-kSubS

We evaluate MCTS-based approaches on INT proofs of length 15. We set $c_{puct} = \tau = 1$ and $\gamma = 0.99$. We tuned $P$ on MCTS (AlphaZero) baseline and we run three variants with $P \in \{5, 15, 50\}$. We run MCTS-kSubS ($k = 3$) with the same set of parameters and with kSubS-specific parameters fixed to $C_2 = C_3 = 4$ (in order to match the setup for corresponding INT BF-kSubS experiments).

We limit the maximum number of actions to $L_a = 24$ for both methods. Having the same number of planning passes $P$, during a single call MCTS-kSubS visits $k$-times more new states than the baseline (because of states visited by the low-level conditional policy). Therefore, to ensure a similar computational budget, we limit the number of planner calls to $L_p = 8$ for MCTS-kSubS and to $L_p = 24$ for the baseline – so the number of states visited over the course of a single solver run is similar for both methods.

Top-left part of Figure 1 illustrates results of MCTS experiments. For every number of planning passes $P$, MCTS-kSubS has significantly higher success rate than the corresponding baseline experiment. The highest difference is $0.52$ for $P = 5$ ($0.88$ for MCTS-kSubS, $0.36$ for the baseline) and slightly decreases with increasing number of passes to still impressive $0.43$ for $P = 50$ ($0.91$ for MCTS-kSubS, $0.48$ for the baseline). Comparing MCTS-kSubS for $P = 5$ with the baseline for $P = 50$, shows advantage of our method still by a significant margin of $0.40$, despite having 10 times smaller computational budget.

MCTS-kSubS performed better also in terms of run time. For every tested $P$ it was at least twice as fast as the corresponding baseline experiment.

High effectiveness of MCTS-kSubS, in terms of both search success rate as well as run time, shows that our kSubS method is not specific to BestFS planner, but potentially allows to boost a wide range of other planners.

# B   Architectures and hyperparameters

## B.1   Transformer

For INT and Rubik we use mBART [28] – one of the state-of-the-art sequence-to-sequence transformer architectures. To speed up training and inference we use its lightweight version. We reduced the dimensionality of the model, so the number of learned parameters decreased from the original 680M to 45M. The set of our hyperparameters matches the values proposed in [49]: we used 6 layers of encoder and 6 layers of decoder; we adjusted model's dimension to 512 and number of attention

heads to 8; the inner-layer of position-wise fully connected networks had dimensionality 2048. The difference in our model's size compared to 65M parameters reported in [49] results from vocabulary size. For our problems, it is enough to have 10-70 distinct tokens, whereas natural language models require a much larger vocabulary (tens of thousands of tokens).

For inference we used number of beams equal to 16 on INT and 32 on Rubik's Cube.

### B.2 Sokoban

In Sokoban, we use three neural network architectures: for generating subgoals, for assigning value and one for baseline policy.

We took the value function network architecture from [29]. For the subgoal generator network we used the same convolutional architecture as in [29], with two exceptions. First, instead of predicting single regression target we predicted distribution over $d \times d \times 7 + 1$ classes. Secondly, we added batch norm layers between convolutional layers to speed up training. To make the comparison between BestFS and BF-kSubS fair, we also evaluated the training of expert from [29] with additional batch norm layers, but it turned out to actually hurt the performance of the expert. The architecture for baseline policy was the same as in [29] with only one modification: it predicts one of our actions instead of a single number.

## C   Data processing and datasets

### C.1   Sokoban

**Dataset**. We collected expert datasets using an RL agent (MCTS-based) from [29]. Precisely, we trained 3 agents on Sokoban boards of different sizes ($12 \times 12$, $16 \times 16$ and $20 \times 20$, all with four boxes). During the training process, we collected all successful trajectories, in the form of sequences of consecutive states. The number of trajectories in our datasets were: 154000 for $12 \times 12$ boards, 45000 for $16 \times 16$ boards and 21500 for $20 \times 20$ boards. The difference comes from the fact that the larger boards take more time to solve, hence fewer data is collected in the same time span.

**Subgoal generation**. For a given `state` the generation of subgoals is depicted in Algorithm 7. We maintain a queue of modified states (MS). Iteratively we take a MS from queue, concatenate it with `state` and pass through subgoal generator network (`subgoal_net`.SORTED_PREDICTIONS). This produces a probability distribution over candidates for further modifications of given MS. We take the most probable candidates, apply each of them to MS, and add the new modified states to the queue. If among the best subgoal generator network predictions there is a special "valid subgoal" token (encoded with $d \times d \times 7 + 1$), we put MS to subgoal candidates list (`subgoals_and_probs`). During this process, each MS is assigned the probability, which is a product of probabilities of modifications, leading to this MS. When the queue is empty, we take subgoal candidates and choose the ones with the highest probability such that the target probability ($C_4$) is reached (similar to Algorithm 3). The generation of subgoals for a given state is illustrated in Figure 5.

This process is designed to be computationally efficient. The majority of subgoals differ from the input by only several pixels, which leads to short paths of point-wise modifications. Note, that we do not utilize any Sokoban-specific assumptions.

**Datapoints for training**. Preparing data points for the training of the generator is described in Algorithm 8. For each trajectory in the dataset, we choose randomly 10% of state pairs for the training (we do not use all states from a trajectory in order to reduce the correlation in the data).

**Low-level conditional policy**. In Algorithm 9 we describe the BFS-based algorithm that verifies subgoals in Sokoban.

**Performance of RL agent**. We observed that each of the three RL agents we used (for $12 \times 12$, $16 \times 16$ and $20 \times 20$ boards), had a significantly lower success rate than our method's counterparts (that learns from these agents). For $12 \times 12$ boards it could solve around 78% of problems, for $16 \times 16$ boards it dropped to 67% and for $20 \times 20$ it was only 60%.

---
**Algorithm 7** Sokoban subgoal generator
---
**Require:**          d          dimension of a board

          internal_cl    a number between 0 and 1

          subgoal_net    CNN returning distribution over modifications.

**function** GENERATE_SUBGOALS(state)

  subgoals_and_probs ← []

  q ← Queue()                                                              ▷ FIFO queue

  q.INSERT((state, 1))

  **while** not q not empty **do**

    modified_state, parent_prob ← q.POP( )

    network_input ← CONCATENATE(state, modified_state)

    predictions, probs ← subgoal_net.SORTED_PREDICTIONS(network_input)

    total_p ← 0

    **for** prediction, p ∈ (predictions, probs) **do**

      **if** total_p ≥ internal_cl **then break**

      total_p ← total_p + p

      **if** prediction = d × d × 7 + 1 **then**

        subgoals_and_probs.ADD((modified_state, parent_prob × p))

      **else**

        new_modified_state ← APPLY_CHANGE(modified_state, prediction)

        q.INSERT((new_modified_state, parent_prob × p))

  subgoals_and_probs ← SORT_BY_PROBABILITY(subgoals_and_probs)

  total_p ← 0

  **for** subgoal, p ∈ subgoals_and_probs **do**

    **if** total_p > $C_4$ **then break**

    subgoals.ADD(state)

    total_p ← total_p + p

  **return** subgoals

**function** APPLY_CHANGE(state, modification)

  ▷ modification is an integer in range [1, d × d × 7] encoding which pixel of state

  ▷ to change (and to which value).

  row ← $\frac{\text{modification}}{\text{d} \times 7}$                                                          ▷ Integer division

  column ← $\frac{\text{modification} - \text{row} \times \text{d} \times 7}{7}$                                  ▷ Integer division

  depth ← modification − row × d × 7 − column × 7

  modified_state ← state

  SET_TO_ZEROES(modified_state[row, column])

  modified_state[row, column, depth] ← 1

  **return** modified_state
---

---

**Algorithm 8** Sokoban generate network inputs and targets

---

**Require:**     d     dimension of a board
  **function** GENERATE_INPUTS_AND_TARGETS(state, subgoal)
      inputs ← []                                                                      ▷ empty list
      targets ← []                                                                     ▷ empty list
      modified_state ← state
      input ← CONCATENATE(state, modified_state)
      inputs.APPEND(input)
      target_class_num ← 0
      **for** $i \in 1 \ldots d$ **do**
         **for** $j \in 1 \ldots d$ **do**
            **for** $c \in 1 \ldots 7$ **do**
               target_class_num ← target_class_num + 1
               **if** subgoal[i, j, c] = 1 AND modified_state[i, j, c] = 0 **then**
                  targets.APPEND(target_class_num)
                  ▷ Numpy notation, replace pixel values on position $(i, j)$ with values
                  ▷ from subgoal
                  modified_state[i, j, :] ← subgoal[i, j, :]
                  input ← CONCATENATE(state, modified_state)
                  inputs.APPEND(input)
      ▷ Last target: no more changes to the modified_state are needed (class enumerated
      ▷ with $d \times d \times 7 + 1$)
      targets.APPEND($d \times d \times 7 + 1$)
      **return** inputs, targets

---

---

**Algorithm 9** BFS low-level conditional policy

---

**Require:**          $k$          limit of steps
             $M$          model of the Sokoban environment
**Use:**          bfs_queue     BFS queue;
                        stores pairs of a state and the action path to it (from the root).
  # Initialize bfs_queue to empty
  **function** GET_PATH($s_0$, subgoal)
      step ← 0
      bfs_queue.ADD$((s_0, []))$
      **while** bfs_queue not empty **do**
         s, action_path ← bfs_queue.POP( )
         **for** action $\in$ action_space **do**
            s ← $M$.NEXT_STATE(s, action)
            action_path.APPEND(action)
            **if** s = subgoal **then**
               **return** action_path
            **if** LEN(action_path) $< k$ **then**
               bfs_queue.ADD$((s, action\_path))$
      **return** []

---

## C.2 INT

**State representation.** A state in INT consists of objectives to prove and ground truth assumptions, which are logic statements that are assumed to hold and can be used in proving. Each objective, as well as each ground truth assumption, is a mathematical statement. In our setup, as in the original paper [55], there is always only one objective to prove, but there may be a varying number of ground truth statements.

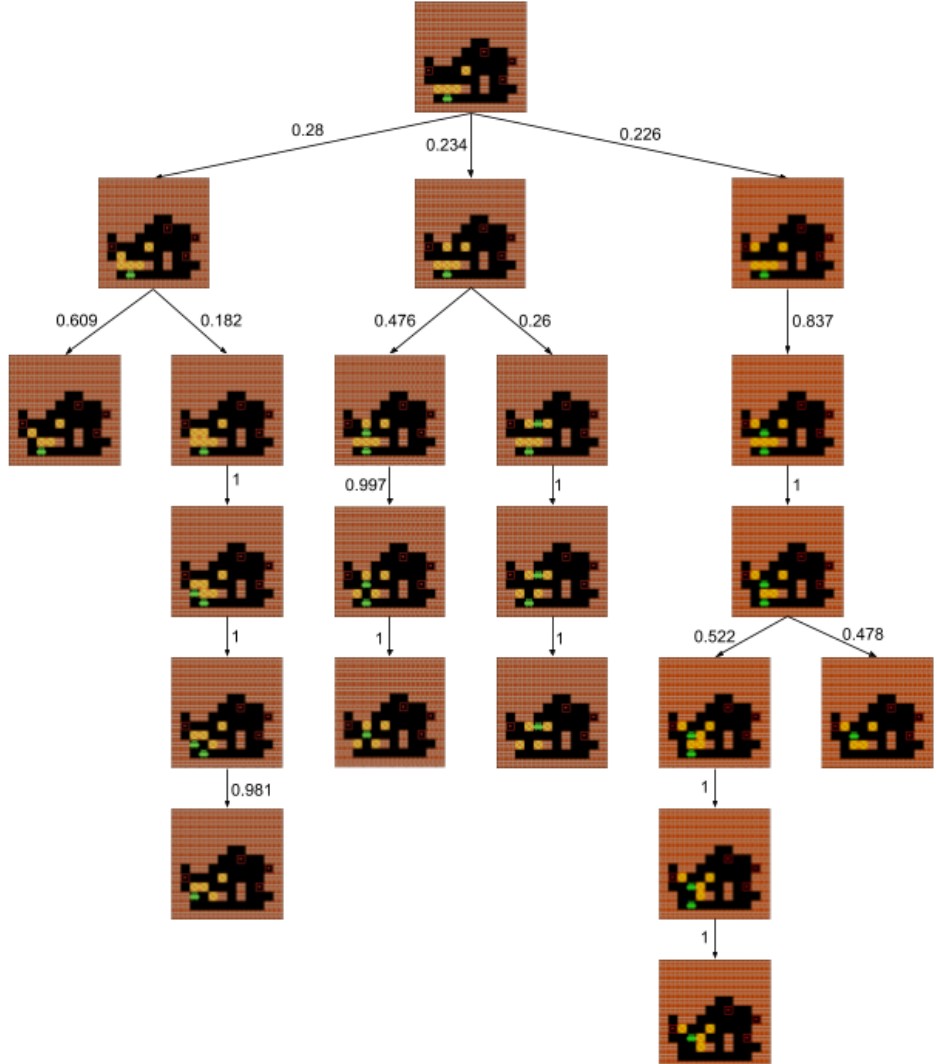

Figure 5: A detailed view of subgoal generation for Sokoban. Arrow represent probabilities of a given modification. Final subgoals are located in the leaves.

Each mathematical statement can be converted to a string by a method `logic_statement_to_seq_string` in INT library. In our code, we represent the full state as a string in the following form (all symbols are tokens, not logical operations):

$$\#[\text{objective}]\&[\text{1st ground truth}]\&[\text{2nd ground truth}]\&\ldots\&[k-\text{th ground truth}]\$$$

For example, the state representation could look like this:

$$\#(((b+b)*((b+b)*(b+b)))*((((b+b)+f)*(b+b))*(b+b)))\geq 0\&(b+f)=b\&(b+f)\geq 0\$$$

**Action representation.** An action in INT consists of a chosen axiom (one from the set of ordered field axioms, see [55, Appendix C]) and a sequence of entities onto which the axiom will be applied. An entity is an algebraic expression that is a part of the larger statement. For example, $(a + b)$ is an entity, which is a part of $(a + b) \cdot c = (1 + f)$. In our code, we represent entities by indicating the symbol of their mathematical operation, or if the entity is atomic (a single variable), by indicating the variable itself. More precisely, directly after the symbol of operation, we add a special character '$\sim$'. For example, we indicate $(a + b)$ inside $(a + b) \cdot c$ in the following way: $(a+ \sim b) \cdot c = (1 + f)$. Typically, in a logic statement there may be several entities that have the same text form, but are

located in different positions, for example $(a + b)$ appears twice in $(a + b) \cdot (a + b) = (1 + 0)$. Our way of encoding actions unambiguously identifies every entity. If the action has more than one input, we use more different indicators.

**Low-level conditional policy input representation.** Low-level conditional policy takes as an input two states: $s$ and $s'$, where $s$ is the initial state and $s'$ is the subgoal. The input is constructed in the following way: first, we represent both $s$ and $s'$ as strings and then we find the difference (character delta) between these strings using the function ndiff from difflib[5] Python library. We observed that using the character delta, instead of the concatenation of $s$ and $s'$, significantly improved the performance.

## C.3  Rubik's Cube

**State representation** The state of the Rubik's Cube is determined by the arrangement of $54$ colored labels on its faces. Therefore, to represent the observations we simply put the labels in a fixed order. An example state is as follows:

$$?byywygrygobbrboorgwbowryoogywbggywrroyrogyowwbrwwbbgg\$,$$

where the tokens *b, g, o, r, w, y* stand for *blue, green, orange, red, white*, and *yellow*. The consecutive blocks of 9 tokens correspond to consecutive faces of the cube. Observe, that not every permutation of colors is valid. For example, the tokens on positions 5, 14, 23, 32, 41, and 50 correspond to centers of faces, thus they are fixed. There are more such constraints, but they are irrelevant to the pipeline itself.

**Action representation** In our experiments we use quarter turns, i.e. an action corresponds to rotating a face by $90°$, either clockwise or counterclockwise. Since the action space contains only 12 elements, we use unique tokens to represent each of them.

**Low-level conditional policy input representation.** The conditional policy takes two states $s$ and $s'$, which correspond to the current state and the state to be reached. To represent such pairs, on every position we put a token corresponding to a pair of colors – one located on that position in $s$ and the other in $s'$. Since there are only 6 distinct colors on the Rubik's Cube, this requires using only 36 tokens.

# D  Training details

## D.1  INT and Rubik's Cube

### D.1.1  Transformer training

For transformer training and inference we used HuggingFace's Transformers library [54]. We did not use any pretrained checkpoints from HuggingFace model hub. We took mBART model class instead – and trained it from scratch in a supervised way using HuggingFace's training pipeline. We generated (or loaded from disk) a fresh dataset for every epoch. Training batch was of size 32. For regularization, we set $dropout = 0.1$, but we did not use label smoothing (as opposed to [49]).

For the Adam optimizer we set $\beta_1 = 0.9$, $\beta_2 = 0.999$ and $\epsilon = 10^{-8}$. Learning rate schedule followed the formula:

$$lr = peak\_lr * \min\left(\frac{step\_num}{warmup\_steps}, \sqrt{\frac{warmup\_steps}{step\_num}}\right),$$

where $peak\_lr = 3 \cdot 10^{-4}$ and $warmup\_steps = 4000$.

The schedule curve matches the one proposed in [49], but they use $peak\_lr \approx 7 \cdot 10^{-4}$.

### D.1.2  Sequence generation

We used beam search with the number of beams set to 16 for INT and to 32 for Rubik's Cube. The number of returned sequences varied from 1 to 4 depending on the network type.

---

[5]https://docs.python.org/3/library/difflib.html

We set softmax temperature to $1$ by default. For Rubik's Cube subgoal generator we tuned this parameter and the value of $0.5$ performed best. We conducted a corresponding experiment for the baseline policy, but the temperature did not impact results in this case, as the policy outputs only a single token. For INT we did not tune the temperature, so we kept the value of $1$.

### D.1.3 Random seeds

Due to use of the supervised learning, we observed little variance with respect to the random initialization. We tested this for the subgoal generator on proofs of length $10$ and for $k = 3$. Namely, we trained $5$ models of the subgoal generator, starting from different initializations. The success rate barely varied, as they stayed in the interval $[0.990, 0.992]$. In the other experiments, we used a single seed.

### D.2 Sokoban

For training of the convolutional networks in Sokoban we set the learning rate to $10^{-4}$ and the number of epochs to $200$.

# E    Wall-time for kSubS

As indicated in Table 2, kSubS builds smaller search graphs. This has the practical advantage of making fewer neural network calls and consequently a substantially better wall-time.

The gains might be as high as 7 times due to costly sequential calls of transformer networks, see Table 5.

| Proof length | 5 | | 10 | | 15 | |
|---|---|---|---|---|---|---|
| Method | BestFS | BF-kSubS (ours) | BestFS | BF-kSubS (ours) | BestFS | BF-kSubS (ours) |
| Total wall-time | 4h 12m | **3h 44m** | 29h 22m | **5h 55m** | 69h 15m | **9h 22m** |
| Avg. generator calls | NA | **3.04** | NA | **3.89** | NA | **6.23** |
| Avg. value calls | 23.34 | **4.01** | 112.35 | **4.80** | 159.46 | **6.59** |
| Avg. policy calls | 22.41 | **8.43** | 112.21 | **13.09** | 161.02 | **20.29** |

Table 5: Resources consumption for INT. We present evaluation on 1000 proofs and split into calls of subgoal generator network (used only in the subgoal search), value network and policy network (we report an average number of calls for a single proof).

# F    Training dataset size analysis

We tested how the success rate of BF-kSubS on 12x12 Sokoban boards depends on the size of the training set. The full dataset consists of 125k trajectories. We trained subgoal generator and value network on subsets consisting of 0.5, 0.25, 0.05 and 0.01 of all trajectories. The results are presented in Table 6.

| Fraction of the dataset | 1 | 0.5 | 0.25 | 0.05 | 0.01 |
|---|---|---|---|---|---|
| Success rate | 0.93 | 0.86 | 0.84 | 0.48 | 0.14 |

Table 6: Sokoban success rates for different training set sizes.

# G    Value errors

## G.1    INT analysis

Due to the size of state spaces, it is impossible to search over the entire space of INT formulas to find the shortest proofs. We instead analyze value estimations along proofs generated by INT engine. The monotonicity analysis in Section 4.6 was performed using 100 such proofs of length 10. The probabilities of value decrease for different step lengths $l$ are presented in Table 7.

## G.2    Sokoban Analysis

Here, we present details related to the Sokoban analysis from Section 4.6. We sampled 500 Sokoban boards (with dimension $(12, 12)$). For each board, we calculated a full state-action graph and minimal distance from each state to the solution (this was needed to compute $S(s)$ sets later on). Since Sokoban graphs can be very large we set the limit on the graph size to 200000, which left us with 119 boards. Next, for each board, we took the sequence of states from the shortest solving path from the initial state (let us call this dataset as *shortest-paths* - SP). For each pair of consecutive states in SP, we calculated the difference of the value estimation, and averaged them, which gave us a mean one-step improvement of 1.34. We calculated this metric for 5 value function networks trained with different initialization, obtaining mean one-step improvement between [1.23, 1.41].

| $l$ | Value decrease prob. |
|---|---|
| 1 | 0.316 |
| 2 | 0.217 |
| 3 | 0.080 |
| 4 | 0.020 |

Table 7

To calculate the standard deviation of value function estimates for $S(s)$ we took SP, and limit it to states $s$ such that $|S(s)| \geq 5$ (lets denote it as SP5). We calculated standard deviation for each

$s \in SP5$ separately. This gave us a mean deviation of $2.43$. (between $[2.24, 2.86]$ for 5 value networks trained with different initialization) The same set SP5 was used to calculate probabilities related to overoptimistic errors on Sokoban described at the end of Section 4.6.

To calculate the above statistics we used the value function trained with supervised learning to approximate the distance to the solution. We observe that similar problems arise also when using value function trained with reinforcement learning [29]. In such setup, mean variance of value function estimates for $S(s)$ is $0.84$, when one step improvement equals to $0.33$ . Probability that there is a state in $S(s)$ with value higher than best immediate neighbour of $s$ is 86% and it drops to 38%, if one considers states closer by $4$ steps.

# H  Example subgoals

## H.1  Example subgoals sets

In this section, we present some example outcomes of the subgoal generator for INT and Sokoban.

### H.1.1  INT

In (1) and (2) we provide two examples of applying the subgoal generator (trained on proofs of length 5) to the given states in INT. The number of subgoals varies since not all of the outputs generated by the network could be reached by the conditional low-level policy.

$$
\begin{aligned}
&\text{Input state: } (((b \cdot b) + ((b + (b+f)) \cdot b)) + (f+f)) \geq ((((b + (b+b)) \cdot b) + 0) + c) \\
&\text{Ground truth 1: } (b+f) = b \\
&\text{Ground truth 2: } (f+f) \geq c \\
&\text{Subgoal 1: } ((b \cdot b) + ((b + (b+f)) \cdot b)) = (((b + (b+b)) \cdot b) + 0) \\
&\text{Subgoal 2: } (((b + (b+f)) \cdot b) + (b \cdot b)) = (((b + (b+b)) \cdot b) + 0) \\
&\text{Subgoal 3: } ((b \cdot b) + ((b + (f+b)) \cdot b)) = (((b + (b+b)) \cdot b) + 0)
\end{aligned}
\tag{1}
$$

$$
\begin{aligned}
&\text{Input state: } ((((b \cdot (\tfrac{1}{b})) + a)^2) + (c + (\tfrac{1}{b}))) = (((((\tfrac{1}{b}) \cdot b) + a) \cdot (a+1)) + c) + (\tfrac{1}{b})) \\
&\text{Subgoal 1: } ((((b \cdot (\tfrac{1}{b})) + a)^2) + (c + (\tfrac{1}{b}))) = ((((b \cdot (\tfrac{1}{b})) + a) \cdot (a+1)) + (c + (\tfrac{1}{b}))) \\
&\text{Subgoal 2: } ((((b \cdot (\tfrac{1}{b})) + a) \cdot ((b \cdot (\tfrac{1}{b})) + a)) + (c + (\tfrac{1}{b}))) = ((((b \cdot (\tfrac{1}{b})) + a) \cdot (a+1)) + (c + (\tfrac{1}{b})))
\end{aligned}
\tag{2}
$$

### H.1.2  Sokoban

Here we present two examples of the outcomes of the subgoal generator trained for $12 \times 12$ boards:



Input board  Subgoal 1  Subgoal 2  Subgoal 3  Subgoal 4

## H.2  Example solutions with subgoals

In this section, we present several examples of solutions obtained with our method. For simplicity, we only show the subgoal states on which the successful trajectories were constructed. In our setup, the last subgoal is always a solution.

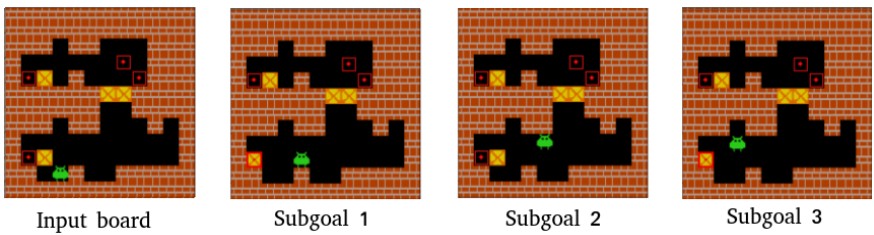

| Input board | Subgoal 1 | Subgoal 2 | Subgoal 3 |

### H.2.1 INT

An example solution of INT problem of length 5:

Problem: $(((0 \cdot ((a+0)+(-(a \cdot 1)))) \cdot (\frac{1}{(0^2)}))+((a+0)+(0^2))) \geq ((((0^2)+(1+(a+0)))+b)+(-((a+0)+f)))$

1st subgoal: $(((0 \cdot ((a+0)+(-(a \cdot 1)))) \cdot (\frac{1}{(0^2)}))+((a+0)+(0^2)))=((0^2)+(1+(a+0)))$

2nd subgoal: $(((0 \cdot ((0+a)+(-(a \cdot 1)))) \cdot (\frac{1}{(0^2)}))+((a+0)+(0^2)))=(1+((a+0)+(0^2)))$

3rd subgoal: $(0+a)=(a \cdot 1)$

4th subgoal: $a=a$

$$(3)$$

### H.2.2 Sokoban

An example solution of Sokoban board:

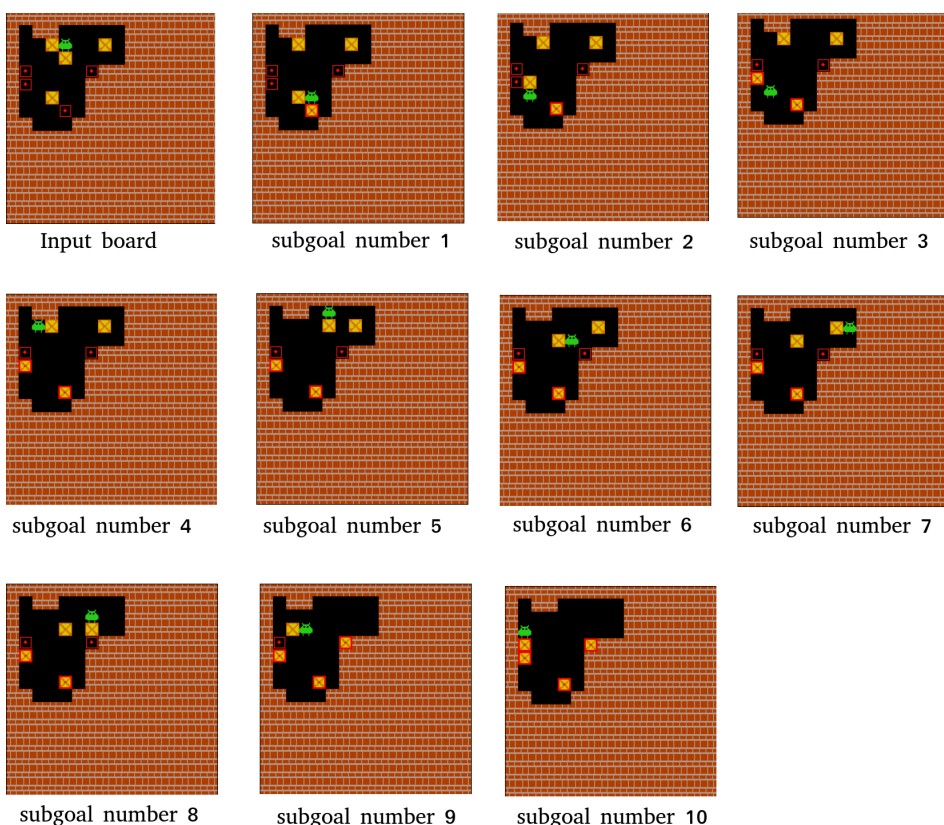

| Input board | subgoal number 1 | subgoal number 2 | subgoal number 3 |

| subgoal number 4 | subgoal number 5 | subgoal number 6 | subgoal number 7 |

| subgoal number 8 | subgoal number 9 | subgoal number 10 |

## I    Baselines

Our first baseline is the low-level policy trained with behavioral cloning from the expert data trained to solve the problem (contrary to the low-level conditional policy $P$, which aims to achieve subgoals). Such policy was used [55]. We verified that our behavioral cloning policy reproduces the results from [55] for proofs of lengths 5.

**MCTS**. As a baseline for MCT-kSubS we used an AlphaZero-based MCTS planner described in Appendix A.1.

**BestFS**. The baseline for BF-kSubS is a low-level planning. We substitute SUB_GENERATE with a function returning adjacent states indicated by the most probable actions of behavioral cloning policy, see Algorithm 10.

---

**Algorithm 10** Low-level generator

**Require:**    $C_3$    number of states to produce
            $NB$    number of beams in sampling
            $\pi_b$    behavioral cloning policy
            $M$    model of the environment

**function** SUB_GENERATE(s)
     actions                         $\leftarrow$
     BEAM_SEARCH$(\pi_b, \text{s}; C_3, NB)$
     subgoals $\leftarrow$ []
     **for** action $\in$ actions **do**
         $s \leftarrow M.\text{NEXT\_STATE}(\text{s}, \text{action})$
         subgoals.APPEND(s)
     **return** subgoals

---

## J   Simple planners

An appropriate search mechanism is an important design element of our method. To show this, we evaluate an alternative, simpler procedure used by e.g. [9] for subgoal-based planning. It works by sampling independent sequences of subgoals and selects the best one. This method solved none of 1000 Rubik's Cube instances despite using the same subgoal-generator as BF-kSubS (which has a success rate of 0.999 with a comparable computational budget).

## K   Investigation of baseline-BestFS on Rubik's Cube

To obtain the training datasets on the Rubik's Cube environment, we generated random paths starting from a solved cube and stored them in the reversed order. These backward solutions are highly sub-optimal: for example, the states obtained by 10 random moves are usually in a distance of about 6 - 7 steps from the final state and the gap gets much larger for a higher number of moves. This means that on collected trajectories only some of the actions indeed lead to the solution and the majority of them only introduce noise.

We observed that the baseline is much weaker than BF-kSubS even for $k = 1$, despite the effort on tuning it. We extended the training of behavioral cloning policy and did a grid-search over parameters to further improve its success rate. We managed to reach no more than 10% success rate for the best version. To provide a meaningful evaluation of the baseline, we also trained it on very short trajectories consisting of 10 random moves. Such a curriculum allowed the behavioral cloning policy to learn, however still suffered from randomness in data (for states located far from the solution).

Interestingly, BF-kSubS for $k = 1$ turned out to perform much better than the baseline. Full understanding of this phenomenon requires additional research, however we hypothesize that in our setup learning to predict states is an easier task that predicting an action. A potential reasons are that: states prediction provides a denser learning signal and Transformers perform better when dealing with sequences (then with predicting a single token).

Note that both kSubS and baseline policy learn from fully off-policy data. The problem of solving the Rubik's Cube is challenging, thus learning from noisy off-policy trajectories can be simply too hard for standard algorithms.

## L   Technical details

### L.1   Infrastructure used

We had 2 types of computational nodes at our disposal, depending on whether a job required GPU or not. GPU tasks used a single Nvidia V100 32GB card (mostly for transformer training) and Nvidia

RTX 2080Ti 11GB (for evaluation) with 4 CPU cores and 16GB RAM. The typical configuration of CPU job was the Intel Xeon E5-2697 2.60GHz processor (28 cores) with 128GB memory.

Each transformer for INT was trained on a single GPU for 3 days (irrespective of proof length and the network type). INT evaluation experiments used a single GPU for a time period varying from several hours to 3 days – with baselines being the bottleneck.

Transformers for Rubik's Cube required more training – every network was trained for 6 days on a single GPU. Because of relatively short sequences representing the cube's state, we were able to run evaluations without GPU. We used 20 CPU cores with 20GB memory, while still fitting in a 3-day run time.

We trained and evaluated Sokoban on CPU only mostly because of rather small neural network sizes. Training time varied from 2 to 3 days, whereas evaluation took only an hour.