# OpenReview forum: "Subgoal Search For Complex Reasoning Tasks"
_NeurIPS.cc/2021/Conference — NeurIPS 2021 Poster_

### Official Review · Reviewer_raVp · 2021-07-17

**Rating:** 7
**Confidence:** 4

**Summary:**

The paper considers the problem of finding a sequence of actions that, starting at an initial state, achieve a goal. The search graph is known in this problem: transition model is known, states are fully observable and actions are deterministic. That means any search algorithm could solve the problem for graphs like Breadth First Search, Best First Search or A*.

The paper proposes to abstract the transition function by mapping a state into a set of steps $k$ steps ahead. When $k=1$, the abstraction corresponds to a subset of the states that can be achieved by applying one action.

The use of lookahead well-established idea in the search community, in different forms, but it seems under-explored for data-driven methods for search. So, the line of work is original.

I celebrate that the paper focuses on deterministic observable known models because this setting allows understanding better the relationship between "search" and "learning". That relationship is a the core of the tension between data-driven and model/reasoning-driven methods.

The research question about the value of predicting $k$ steps ahead is interesting. On the one hand, data-driven methods are combined with search because they perform better than following predictions greedily. AlphaZero is "exhibit A". On the other hand, bounded-length predictions are more likely to remain within the practical scope of data-driven methods.

I do have a serious problem with the use of the word "**analog**" across the paper. I think we cannot write papers about "analog of human intuition" because "analog" implies similarity or comparison. It's a very strong word. I could accept "inspired by an aspect of human intuition", but I find it unacceptable because it sounds like we know what human intuition is. I might be wrong. If that's the case, please justify that using the relevant literature. Otherwise, please just state how the manuscript would be revised so this expressions have a lower tone.

The findings are very interesting. Fig 2 is the most insightful.

However, I think the manuscript have some confusions that make the contributions less clear.
Only three domains are considered. Two use the same setting with a shared family of hyperparameters. A third one uses a different setting.


**Main Review:**

I am confused by the wording of the contributions. Therefore, let me summarize them using the same numbers as in the paper.

1. Subgoal Search using transformer + search algorithm is strong.
2. Same transformer per se is strong.
3. The value functions used in the paper are weak.

The main issue is that they confound the algorithms used with the research question. The 3rd contribution is not about *value functions* in general but about the value function used in the paper. The other two contributions get diluted by

* Using different setup in Sokoban vs the other two domains.
* The lack of comparison with search algorithms.

A minor weakness is

* Given the effort devoted to tuning the algorithms, it becomes relevant to try planning algorithms suitable for this deterministic setting.

This is NeurIPS, so this illustration might be pertinent for this setting, deterministic and where the model is known.

* If the states are atomic, then both search and goal-conditioned RL solves the problem of finding a sequence of actions.
* A state space is factored when any state can be described a set of factors, like predicates+objects. For instance, for sokoban, it could be agent-at(1, 3), block(3,4), etc. In this setting, classical planners find a solution. They receive a high-level description of the model or a simulator like an OpenAI gym.

The evaluation is done in three benchmarks:

* Rubik's cube. Well studied in the search community.
* Sokoban. Well studied in the search and planning communities.

A simulator-based planner is compatible with this setting, so a strong paper must compare with it. I won't ask for that in a conference submission, but planning algorithms cannot be ignored.

REQUIRED: in a related work section, please comment on these two lines of work:

* Classical Planning in Deep Latent Space: Bridging the Subsymbolic-Symbolic Boundary. Masataro Asai, Alex Fukunaga. This one is more flexible than required because it can deal with raw signals.
* Frances, Guillem, et al. "Purely declarative action descriptions are overrated: Classical planning with simulators." IJCAI 2017. This one requires a factored state. They test with Sokoban.

However, the research question is compatible with a simple question that would make the contribution aligned with the research question: is "k step prediction making the difference?"




REQUIRED:
* Please clarify what if BestFS is and what it shares with BF-kSubs.
* Is BestFS equivalent to "k=1"?

REQUIRED: Please make sure the manuscript or your comments report experiment results for the following variations. (They might already be in the paper, depending on what is BestFS implementing).

1. Other values of $k$ for the other two domains besides Sokoban. Otherwise, this statement of page 3 is misleading: "We show, perhaps surprisingly, that this simple objective is an efﬁcient way to improve planning, even for small values of k, i.e. $k \in \{ 2, 3, 4, 5 \}$".
2. When possible, report complete results using Dijkstra. (Sect 4.5 implies part of the work is done)
3. $k=1$
4. $k=1$ and algorithm 3, sub_generate, generates all the states. That is setting $C_3 = C_4 = \infty$. Adjust the sokoban generation B.2 accordingly. In this scenario, get_path should always return the action that is possible. Given that the actions are coming from the policy $\pi$, I don't take that for granted.

This possible lack of information on the points 3 and 4 is the weakest aspect of the paper, as that's the main research question.

REQUIRED:
* Please focus 3rd contribution on the observation about value errors. The importance of subgoal search is covered in the other two contributions.

----

Thank you for your comments. I let my comments in the discussion. Just updating the scores.

**Time Spent Reviewing:**

6

---

> ### Author Response · Authors · 2021-08-10
> **Answer to the reviewer raVp**
>
> We thank the reviewer for valuable comments on our work. We discuss them below and indicate changes that will be made to the camera-ready version. For the sake of brevity, we keep our answers compact. If there are any further questions, we will be happy to provide additional information during the discussion period.
>
> We used the word ‘analog’ in an intuitive and colloquial way, merely to make the narrative more colorful and make our intuitions more relatable. We understand (and agree) with your point that this might be misleading and will remove this word from the text.
>
> We understand some confusion in the statement of our contributions. To make it more clear, we propose:
> to remove ‘transformer’ from point 1.
> keep point 2. We argue this is a valid point since neural architectures are often an integral part of ml-based solutions. To the best of our knowledge, it is the first time that a transformer is used in the context of modeling future states. The previous work, listed in the related work, concentrates on VAE or GAN approaches.
> as to point 3. Errors of a learned value function are a very common and important problem in RL. In the context of search, see e.g. [2] which points out that MCTS backups in AlphaZero allow to average-out approximation errors of the learned value function (see appendix, section "MCTS and Alpha-Beta Search"). Arguably such errors are unavoidable when using deep learning. Following that understanding, their influence might be beneficial for designing data-driven planning methods. We will clarify the camera-ready version and plan to extend the discussion in Sec 4.6.
>
> We acknowledge the review’s concern about the relevance of point 3. To make the case stronger, we will prepare additional analyses with more seeds, neural architectures, and datasets.
>
> A different implementation on Sokoban was meant to illustrate the flexibility of our subgoal approach with respect to various implementation details. We consider this to be a valid and important point. We understand that this, indeed, may feel at odds with the contributions statement. To make it complete, we will prepare BF-kSubS experiments for Sokoban with the same setup as for other domains for the camera-ready version.
>
> Concerning the setting of this work, we propose a simulation-based planner for deterministic domains. We build on classical planning methods developed in the context of atomic states (i.e. BestFS), at the same time we do use the structure of states to facilitate learning. We make no explicit assumption about this structure. For example, the input in INT is a textual formula, which is not a factored representation (at least not in a sense of Sec 2.4.4 of [3]). This explanation will be integrated into the camera-ready version along with the suggested references. We note that the crucial difference is that they do not use hierarchical abstraction (i.e. subgoals). In future research, it might be possible to combine our approach with such methods. For example, we would consider promising using k-step predictions in a factored space.
>
>
> *$k$ is indeed a crucial hyperparameter of our method*. We have run additional experiments to quantify this (see the tables below, for Rubik, we have not run a full 6-day training, curves similar to Fig 2 with full results will be included in the paper). We note that BF-kSubS with $k=1$ is very similar to BestFS. In the cases of small action spaces, they may coincide. For big action spaces, say INT, only a subset of successor states can be visited. For BestFS, a neural policy is used to predict actions, then mapped to states (using the model). BF-kSubS with $k=1$ uses a predictor which directly produces states. On INT the two methods obtain success rates within the estimation error range (0.44 for BestFS; 0.47 for BF-kSubS with $k=1$)
>
> | k (INT)      | 1    | 2    | 3    | 4    |
> |--------------|------|------|------|------|
> | Success rate | ~44% | ~87% | ~91% | ~80% |
>
> | k (Rubik, shorter training)    | 1    | 2    | 3    | 4    |
> |--------------|------|------|------|------|
> | Success rate | ~33% | ~96% | ~99% | ~98% |
>
> Importantly, we observe that performance for $k>1$ is *significantly higher*.
>
> Varying $C_3$ and $C_4$ is also an important aspect. In particular, setting $C_3=C_4=+\infty$ is the setup of the Sokoban baseline (more precisely, $C_3$ is set to the number of actions and $C_4$ is irrelevant). For Rubik’s cube, we tuned $C_3$ (separately for BF-kSubS and BestFS). Increasing the value of $C_3$ does not help; setting it to infinity deteriorates performance to 0. On INT, limiting the number of possible actions is unavoidable as the action space typically consists of millions of possible transformations (especially for long formulas).
>
> We have run additional experiments using the Dijkstra algorithm on 12x12 Sokoban boards. This method turned out impractical due to the computational complexity of the algorithm: $O(|E| + |V| \log|V|)$. We verified that Dijkstra needs to visit typically around a million nodes before finding the first solution, taking considerably longer time than kSubS. In Rubik’s Cube and INT state spaces are too large to perform a search without heuristic. For example, the memory demands for breadth-first-search exceeds our resources (128GB). In fact, for Rubik, [1] showed that bidirectional search could be used but required 5TB external memory and hours of extensive parallel computations.
>
> [1] Sturtevant and Chen. External memory bidirectional search. IJCAI  2016.
>
> [2] Silver et al. "Mastering chess and shogi by self-play with a general reinforcement learning algorithm."  arXiv:1712.01815 (2017).
>
> [3] Russel, Norvig. Artificial intelligence. A modern approach. Third edition, 2010.

---

> > ### Comment · Reviewer_raVp · 2021-08-18
> > **Thank you for your comments**
> >
> > Thank you for your comments. They help me understand better the submission. I am more positive now.
> >
> > Thank you for receiving my point on "analogue". Now you know that such expressions make the work less relatable for some readers.
> >
> > I still think the experimental setting is rather complicated. I think other reviewers are confused too. I'd feel better if this were a journal submission and we can do a quick iteration on the content and the paper organization. Sketches of changes are not the same as the paper done.
> >
> > About the 3rd contribution. I think we still disagree. Let me use a generalization:
> > The paper says "We study the negative influence of XYZ errors on planning and show that using
> > 73 subgoal planning might mitigate this problem". In the response, the authors say: " Errors of XYZ are a very common and important problem in RL.".
> > I agree that errors of XYZ are very common. The point is that the paper is not studying all the XYZ but one in particular, so it´s hard to generalize. For instance, I don't know if there was a small error in the implementation.
> > This is in general a problem with negative results in computer science: is far easier to show that ABC can be done than to prove ABC doesn't work.
> >
> > So, I propose a similar statement that is coherent with the rest of the paper. (I'll try to stay close to the manuscript):
> > Replace
> > "3. We study the negative influence of value function errors on planning and show that using
> > 73 subgoal planning might mitigate this problem."
> > by
> > "3. We show that using subgoal planning help to mitigate *in our experiments* the negative influence of our value function errors on planning"
> >
> > Adding "*in our experiments*" allows keeping the negative state within the experiments without general claims about that value function errors. We don't know if next year a slightly different definition of error could solve this issue. If the paper gets accepted and the 3rd contribution is as it is, an archived paper would contradict new findings without intention. I'm trying to avoid that.
> >
> > Point taken about the factored space. Please say so explicitly in the paper.
> >
> > I do think the point about factored representations deserves to be mentioned. They offer a ceiling for the performance. Longer-term, it might be possible that some end-to-end method discovers the right latent structure and matches that performance.
> >
> > *** So, please comment explicitly about the performance of those other methods even if they are uncomparable. A few sentences would do.
> >
> > ----
> >
> > Thank you for the experiments. Clarify the limitation of your experiments to offer additional confidence to the reader.
> >
> > One million nodes is not really that much for these problems. Just open a recent paper accepted the SOCS conference.
> > Richard Korf reported for Rubik´s cube:
> > "Complete searches to depth 16 require an average of 9.5 billion nodes, and take less than four hours."
> > That was in 1997. https://www.aaai.org/Papers/AAAI/1997/AAAI97-109.pdf
> >
> > ----
> >
> > In summary, the paper is onto something. I'm ready to let pass the confusion. We should accept the paper.

---

> > > ### Author Response · Authors · 2021-08-19
> > > **Answer to the reviewer raVp**
> > >
> > > We again thank the reviewer for useful and constructive comments, which, we believe, will improve our paper.
> > >
> > > Concerning the 3rd contribution. We were reluctant to remove this contribution as we argue that the intersection of learning and planning with emphasis on error analysis is an important area. Having said that, indeed, our results are of a negative flavor and as you pointed out this requires extra care in stating limitations. We will follow your proposition to add “in our experiments” phrase (and review the whole noise section carefully). We thank you for your suggestion.
> > >
> > > As to the other points, we will add a discussion about the factored spaces and related methods. Moreover, we will add the additional experiments, stating clearly their limitations. These changes, we believe, will provide a broader context for our work.

---

### Official Review · Reviewer_t8oJ · 2021-07-17

**Rating:** 7
**Confidence:** 2

**Summary:**

This paper developed a subgoal search approach, called kSubS, for identifying subgoals of complex reasoning tasks. Leveraging the developed search approach, a few systems were developed, including two that were referred to as MCTS-kSubS and BF-kSubS in the paper. The systems were tested using domains of Sokoban, Rubik's cube and INT. Results showed that the developed system outperformed baselines of AlphaZero and BestFS.

**Limitations And Societal Impact:**

Adequate.

**Main Review:**

Overall, the paper is technically sound, and the results to some extent showed the effectiveness of the proposed methods. The three benchmark problems are well selected. There are a few issues.

It's believed that the developed approaches' performance highly depends on the quality of the collected expert data. To make a strong case, it's necessary to compare the proposed methods to those from the literature that also leverage expert data, and show that the proposed methods require less data, are robust to low-quality data, or can better utilize the data to improve the search process.

It's unclear from the paper how SubS and kSubS (used interchangeably in the paper) are different from each other.

There is the following 2021 paper that learns imagined subgoals for RL. Since the paper is very new, it's not a problem of not citing the paper -- just a reference.  Goal-Conditioned Reinforcement Learning with Imagined Subgoals, ICML 2021   Also, the reviewer is curious if possible the developed subgoal search methods can be applied to RL-based problem solving methods. For instance, the following paper showed that RL methods can be effective to Rubik's cube (one of the three domains of this paper) as well.  "Solving the Rubik's Cube with Approximate Policy Iteration". ICLR 2018

It's not evident that how to utilize training data from experts to speed up the search process is a non-trivial problem. Intuitively, the subgoals decompose a complex problem into subproblems that are easier, c.f., the original problem. So the idea is quite simple, which is not a problem by itself. But the reviewer is curious if there are more competitive baselines from the literature that can be included to improve the experiment section. If this is really the first work of using supervised learning methods to identify subgoals for problem decomposition, then it's necessary to make it clear in the paper.

**Time Spent Reviewing:**

1.5

---

> ### Author Response · Authors · 2021-08-10
> **Answer to the reviewer t8oJ**
>
> We thank the reviewer for his/her comments.
>
> When it comes to baselines using expert data, Figure 3 compares BF-kSubS with standard behavioral cloning agent (the first two bars show performance on INT proofs of length 10 which does not require out-of-distribution generalization). The ‘no-search’ baseline has much worse results than our method (0.73 vs 0.99 success rate). Behavioral cloning on Rubik fails to solve any cube. Stronger imitation learning methods have additional requirements (e.g., Dagger requires a querying expert during training). For these reasons, we believe that the baselines used in our work (standard search algorithms based on networks trained on the same expert data as kSubS) are appropriate and fairly strong. Concerning data quality, we point out an interesting ablation in the amount of expert data (suggested by Rev RAfv; see the result table there).
>
> Our algorithm is the first successful hierarchical planner for combinatorially complex domains to the best of our knowledge. There is a lot of work in subgoal-based planning in the domains with visual input and continuous control [17, 23, 25, 22, 15, 7, 26]. They are based on variational architectures, which map observations into latent states. These methods are complex, and adjusting them to work in our settings is a task for a separate research project. The closest ones: [9] and [24] have additional requirements that make comparison impossible or unfair. [9] requires access to expert-provided subgoal-predicate functions. Moreover, it performs an exhaustive local search, which would be impractical for Rubik and impossible for INT. [24] requires access to the goal-state and a small state-space (or expert-designed function mapping states to smaller space of subgoals).
>
> The application of our method for Reinforcement Learning agents is indeed an exciting research direction. We believe (based on out-of-distribution generalization and improving over expert performance) that our method has the potential to be put in a self-improving loop, as listed in Sec 5. We leave this for future work as this likely will require additional algorithmic developments.
>
> We thank the reviewer for pointing us to Goal-Conditioned Reinforcement Learning with Imagined Subgoals. We will add this reference to related work.

---

> > ### Comment · Reviewer_t8oJ · 2021-09-03
> > **Thank you**
> >
> > Thank you for the response and being receptive. My overall evaluation remains positive.

---

### Official Review · Reviewer_txcJ · 2021-07-18

**Rating:** 7
**Confidence:** 3

**Summary:**

The paper introduces a subgoal generator to accelerate the search
process to solve a given task.

**Limitations And Societal Impact:**

Limitations appropriately discussed.

At this stage of discussion, I do not consider the Societal Impact discussion relevant to this paper.

**Main Review:**

The present paper introduces an algorithm that uses a
subgoal-generator to generate subgoals which are then driven to by a
local policy, and demonstrates its use in the context of Sokoban,
Rubik's Cube, and INT (theorem prover).

The algorithm is relatively straightforward and simple to implement
and shows good improvement compared to existing methodologies.

The reviewer found it not obvious to understand how the subgoal
generator generates its subgoals. What does the prediction refer to?
Where does the reference come from? There is some implicit mentioning
of expert training, however, that means that the algorithm needs some
seeding. More discussion is now needed to make clear what the
advantage of using such an algorithm is. It is mentioned that there is
some out-of-distribution generalization ability, which is probably the
most interesting aspect of the study, but here more details as to how
that emerges/works might have been useful in addition to experimental
benchmarking, especially in a conference such as NeurIPS.

- line 131: it is crucial for the understanding of the algorithm to
  explain what is being predicted. There is only some mentioning of
  expert data collection, but this is the core part of the algorithm
  and needs to be more detailed. Note that Algorithm 3 is not much
  more illuminating - what does the Beam_Search do here, and what does
  the probabilities bookkeeping do?
- Footnote 3: AlphaZero spelling
- Fig. 4: "increase the distance" - I would treat "12%" as a plural


**Time Spent Reviewing:**

2

---

> ### Author Response · Authors · 2021-08-10
> **Answer to the reviewer txcJ**
>
> We thank the reviewer for his/her comments.
>
> kSubS indeed requires expert data, which we make explicit in the introduction and methods. We will make this more prominent. We note that the training in the planner-learner loop (akin to AlphaZero) can be possibly developed (see Sec 5), thus, dropping the dependence on the expert input. Details of expert data collection are available in Appendix (Section C.1, Section J).
>
> The main advantage of our algorithm is that it is a well-performing planner. We stress that it goes (sometimes much) above the expert data (note that we allow this data to be low quality). In this respect, we also point out an interesting ablation in the amount of expert data (suggested by Rev RAfv; the table there). In the broader view, efficient learning of temporal abstraction is considered to be an important milestone on the way to general intelligence [1]. This is closely connected to out-of-distribution generalization. Following [2], we speculate that search is a computational mechanism, which delivers OOD. Interestingly, this effect is stronger when using subgoals. We think it is a prominent research direction and will put it explicitly in the future work section.
>
> Regarding Algorithm 3, we see the need to improve its presentation (this point was also raised by Rev #RAfv). Perhaps Beam_Search should be removed; it is a standard technique when seq2seq models are used (transformer in our case) but is not essential to our method.
>
> [1] Sutton, Precup, Singh. "Between MDPs and semi-MDPs: A framework for temporal abstraction in reinforcement learning." Artificial intelligence 1999.
>
> [2] Wu, Yuhuai and Jiang, Albert and Ba, Jimmy and Grosse, Roger “INT: An Inequality Benchmark for Evaluating Generalization in Theorem Proving”, ICLR 2021

---

> > ### Comment · Reviewer_txcJ · 2021-08-18
> > **Response 1**
> >
> > Point about possibility of generating expert input via planner-learner loop taken.
> >
> > The point of temporal abstraction being a milestone towards general intelligence is quite generic and not really an answer. This is well known. The question is why it would work so well on a such intricate benchmark such as INT. It's not like subgoal/hypothesis generation hadn't been tried before. If there is any idea why it works better than other algorithms, this should enter the paper. The reviewer found the comments to RAvf not entirely clear in that respect.
> >
> > If BeamSearch is not essential, it can be removed (Reviewer RAvf also asked about it). If it is left in, it should be explained *what* it is doing - not how. The reviewer knows how BeamSearch works. They just do not understand what it is actually searching for here.
> >
> > What is OOD?

---

> > > ### Author Response · Authors · 2021-08-19
> > > **Answer to the reviewer txcJ**
> > >
> > > Thank you for your comments. They are helpful feedback to us.
> > >
> > > We agree that finding an explanation of the effectiveness of our method is an important and interesting question. The general idea has been known and explored for a long time indeed. We argue that the value of our work comes from finding a combination of neural net/transformer-based learning, the definition of subgoals, and the choice of planners, which is effective in practice. It is the first time such an approach was successfully applied to combinatorially complex domains to our best knowledge. We leave finding a more detailed explanation for future work (our experiments in Sec 4.4 and Sec 4.5 are meant to be the very first steps toward this goal).
> > >
> > > We treat beam search (BS) as a tool to achieve an important objective, namely to sample subgoals that have high likelihood. We admit that the current description (having it only in Alg 3) is indeed confusing. We propose to amend this by adding an explanation in the method section. An alternative of removing it from the main text is also a viable option. In our view, though, it may harm the completeness of exposition (additionally taking into account that BS  induces additional hyperparameters, used explicitly in Table 1).
> > >
> > > We should have been clear about abbreviating out-of-distribution (OOD) generalization. Repeating, our results support the hypothesis of [1] that search itself is an OOD mechanism. We will mention this explicitly and further extend the discussion to compare it with the related claims in [2].
> > >
> > > [1] Wu, Y., Jiang, A., Ba, J., Grosse, R. “INT: An Inequality Benchmark for Evaluating Generalization in Theorem Proving”, ICLR 2021
> > >
> > > [2] Hamrick at al., On the role of planning in model-based reinforcement learning, ICLR 2021

---

> > > > ### Comment · Reviewer_txcJ · 2021-08-26
> > > > **Final comments**
> > > >
> > > > I am happy for you to amend the beam search explanation - I have no issues with it staying in, no need to remove it, just explain what precisely it's searching for.
> > > >
> > > > If you add the further clarifications you mention, this should be fine with me.

---

### Official Review · Reviewer_RAfv · 2021-07-20

**Rating:** 7
**Confidence:** 4

**Summary:**

This work proposes a method for search, considering the transition model is given. The proposed framework works as follows: given a state, a subgoal predictor is used to predict subgoals k-steps ahead. A policy network is used to get the actions to reach those subgoals and a value function is used to judge the value or benefit of being in a state. All of these are trained on an expert set of demos. Empirical results show that this algorithm performs better than BestFS or MTCS.

**Limitations And Societal Impact:**

It is not clear to me why adding a subgoal predictor would help. My thought is that it should have been inherently learned by the network. Also, comparisons/discussions with some subgoal-based methods are missing.

**Main Review:**

- I understand that the Low-level policy, value function, and subgoal predictor are learned using expert data. The subgoal is trained to predict the state k-steps from now. Is the predictor deterministic? How do you get multiple subgoals for a single current state. Also, what is the beam-search in Algorithm 3 optimizing?

- If I have to compare with just BestFS or MCTS at a very high level, then the proposed method basically breaks down the next action into two parts - generate subgoals, and then get actions to go those subgoals. It is not clear to me exactly why this is helping the search process? This is mentioned in Line 131, but with no solid intuition behind it.

- Subgoals have been studied in the past for policy learning [A] and efficient imitation learning [B], except, the proposed method is applied to a special case of the settings considered in these works. The authors should discuss and compare the novelty w.r.t. these works.

- What is the sensitivity of the proposed method to the number of expert demos?

[A] Automatic Goal Generation for Reinforcement Learning Agents, ICML 2018
[B] Learning from Trajectories via Subgoal Discovery, NeuIPS 2019

**Time Spent Reviewing:**

2

---

> ### Author Response · Authors · 2021-08-10
> **Answer to the reviewer RAfv**
>
> We thank the reviewer for his/her comments.
>
> The transformer model that we use, is capable of modeling probability distributions over sequences. In our case, the sequences are the states (for example, textual formulas in INT).  Beam search is a (commonly used) algorithm that generates high likelihood samples for this distribution. A subtle issue is, what probability distribution is learned from the data given the fact that we train on (semi)-deterministic dataset. We speculate that this follows from ambiguities of the dataset and generalization power of neural nets (similarly to the language modeling), however, this needs to be investigated more deeply. From the practical perspective, we check that the transformer model generates diverse subgoals (as indicated in Section 4.5; examples can be found in Appendix G.1.1). The above discussion will be extended and included in the camera-ready version.
>
> Thank you for pointing out additional papers. We have carefully reviewed them and will include them in the camera-ready version. The main difference between our work and [A, B] is that we use planning. This is in line with our focus to tackle combinatorially hard problems, which arguably cannot be solved with a pure neural network approach. Another technical, but important, the difference is that subgoals must be predicted precisely. For example, almost any random change to the INT formula makes it meaningless. This point is so critical, that we make additional verification in Algorithm 2.
> We will present more arguments why predictor would help using an additional page available in the camera-ready version. A high-level answer might be that using the subgoals reduces the distance to the goal k-fold. For hard problems, this is a substantial gain, which outweighs the costs (e.g. of having to construct the solution path). Such arguments are in line with general explanations, which see searching over subgoal space as operating on higher temporal abstraction and might be considered as a crucial tool of intelligence [1,2] (and lead to recent successful results in Hierarchical Reinforcement Learning [3, 4]). To conclude, we note in this work we mainly concentrate on showing that the method indeed works, leaving deep understanding to further research.
> Following your request, we did additional experiments to investigate how the number of expert demonstrations affects our method. We trained the subgoal generator for Sokoban (12x12) on 5 different numbers of expert trajectories.
>
> | No. trajectories | 1k  | 5k   | 25k  | 125k | 250k |
> |------------------|-----|------|------|------|------|
> | Success rate     | ~1% | ~15% | ~40% | ~90% | ~95% |
>
> We see that a modest number of trajectories is enough to get non-trivial performance. These results along with a discussion will be included in the camera-ready version.
>
> [1] R. Sutton, D. Precup, S. Singh. "Between MDPs and semi-MDPs: A framework for temporal abstraction in reinforcement learning." Artificial intelligence 112.1-2 (1999)
>
> [2] G. Drescher Made-up minds: a constructivist approach to artificial intelligence. MIT press, 1991.
>
> [3] Nachum, et al. "Data-Efficient Hierarchical Reinforcement Learning." Advances in Neural Information Processing Systems 31 (2018)
> [4] Sharma et al. "Dynamics-Aware Unsupervised Discovery of Skills." International Conference on Learning Representations. 2019.

---

> > ### Comment · Reviewer_RAfv · 2021-08-25
> > **Response**
> >
> > I am still not fully convinced about intuition, other than the fact that it helps empirically. Some toy experiments to understand it better will be a good step. If would be good to see such experiments/analysis in the updated version of the work paper.

---

> > > ### Author Response · Authors · 2021-08-27
> > > **Answer to the reviewer RAfv**
> > >
> > > Thank you for your suggestion. We agree that further analysis would be valuable; this might be indeed the case to find some well-isolated small experiments.
> > >
> > > Let us consider the following grid-world setup. Let the state space be  $S=${$1, …, n$}$^m$, with some $s_g \in S$ being the goal state. We assume the cost of 1 for each move, thus the perfect value function $V_p$ would be the (negative) distance to the goal.
> > >
> > > We assume that the perfect value function is not available. Instead, one has access to its ‘noisy version’ $V_n$.
> > >
> > > We would analyze the behavior of the following methods:
> > >  * **BestFS**: expanding state $s$ adds to the queue $C_3$ neighbors of $s$ with the highest value.
> > >  * **BF-kSubS**: expanding state $s$ adds to queue $C_3$ states *within distance* $k$ of $s$ with the highest value.
> > >
> > >
> > > In this simplified setting, we aim to validate our intuitions on how k-step subgoals alleviate the effect of imperfections of $V_n$. The simplest model is to put $V_n = V_p +$ i.i.d. noise; correlated noise is another natural example. Operationally, we will evaluate the performance (number of steps to reach the goal state) in numerical experiments. We hope that theoretical analysis based on random walk theory [1] would also be possible in some cases.
> > >
> > > [1] Lawler, Gregory F., and Vlada Limic. Random walk: a modern introduction. Vol. 123. Cambridge University Press, 2010.

---

### Decision · Program_Chairs · 2021-09-27

**Decision:**

Accept (Poster)

**Comment:**

This paper presents an interesting approach to problems with long decision horizons, by generating and planning to subgoals on the path to the full solution.
After a discussion largely around clarifying some points in the paper, the reviewers all felt that this paper should be accepted.
The authors should update the descriptions in the paper as suggested by the reviewers.